# Autoregressive Video Generation without Vector Quantization

**Haoge Deng**[1,5*], **Ting Pan**[2,3,5*], **Haiwen Diao**[4,5*], **Zhengxiong Luo**[5*], **Yufeng Cui**[5],
**Huchuan Lu**[4], **Shiguang Shan**[2,3], **Yonggang Qi**[1†], **Xinlong Wang**[5†]

[1]Beijing University of Posts and Telecommunications
[2]Key Laboratory of Intelligent Information Processing, ICT, CAS
[3]University of Chinese Academy of Sciences
[4]Dalian University of Technology    [5]Beijing Academy of Artificial Intelligence

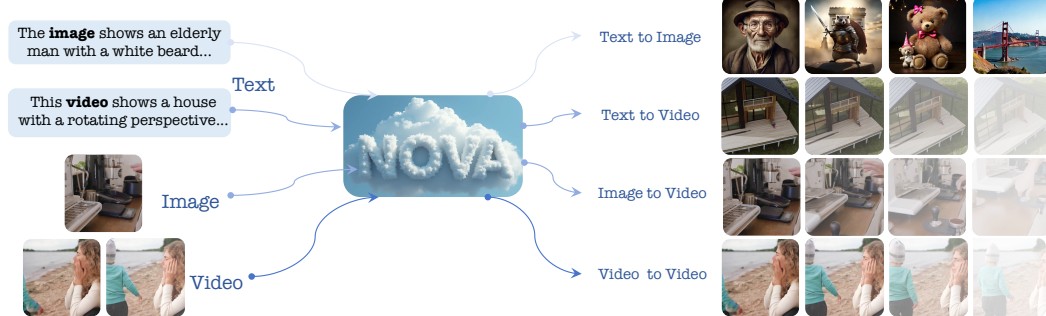

Figure 1: **NOVA** is a non-quantized autoregressive model for efficient and flexible visual generation.

## Abstract

This paper presents a novel approach that enables autoregressive video generation with high efficiency. We propose to reformulate the video generation problem as a non-quantized autoregressive modeling of temporal *frame-by-frame* prediction and spatial *set-by-set* prediction. Unlike raster-scan prediction in prior autoregressive models or joint distribution modeling of fixed-length tokens in diffusion models, our approach maintains the causal property of GPT-style models for flexible in-context capabilities, while leveraging bidirectional modeling within individual frames for efficiency. With the proposed approach, we train a novel video autoregressive model without vector quantization, termed **NOVA**. Our results demonstrate that **NOVA** surpasses prior autoregressive video models in data efficiency, inference speed, visual fidelity, and video fluency, even with a much smaller model capacity, *i.e.*, 0.6B parameters. **NOVA** also outperforms state-of-the-art image diffusion models in text-to-image generation tasks, with a significantly lower training cost. Additionally, **NOVA** generalizes well across extended video durations and enables diverse zero-shot applications in one unified model. Code and models are publicly available at https://github.com/baaivision/NOVA.

## 1 Introduction

Autoregressive large language models (LLMs) (Brown et al. (2020); Touvron et al. (2023)) have become a foundational architecture in natural language processing (NLP), exhibiting emerging capabilities in in-context learning and long-context reasoning. In autoregressive (AR) vision generation domain, prior approaches (Ramesh et al. (2021); Ding et al. (2021); Yu et al. (2022); Yan et al. (2021); Villegas et al. (2022); Kondratyuk et al. (2023); Wang et al. (2024a)) typically transform images or video clips into a discrete-valued token space using vector quantization (Van Den Oord et al. (2017);

---

*Equal Contribution. This work was done when the first three authors were interns at Beijing Academy of Artificial Intelligence. †Corresponding Author: *wangxinlong@baai.ac.cn, qiyg@bupt.edu.cn*

Esser et al. (2021)), which are then flattened into sequences for token-by-token prediction. However, it is challenging for vector-quantized tokenizers to achieve high fidelity and high compression simultaneously. More tokens are required for a high quality. Thus, the cost increases substantially with higher image resolutions or longer video sequences.

In contrast, video diffusion models (Brooks et al. (2024); Kuaishou (2024); Blattmann et al. (2023)) learn with highly compressed video sequences in a compact continuous latent space. However, most of them only learn the joint distribution of fixed-length frames, lacking the flexibility to generate videos with varied lengths. More importantly, they do not possess the in-context abilities of autoregressive models, *i.e.*, solving diverse tasks in context with a unified model such as GPT for language.

In this work, we present **NOVA**, which addresses the issues above and enables autoregressive video generation with high efficiency. We propose to reformulate the video generation problem as a non-quantized autoregressive modeling of temporal *frame-by-frame* prediction and spatial *set-by-set* prediction. **NOVA** is inspired by Emu3 (Wang et al. (2024a)) for autoregressive video and multimodal generation, and MAR (Li et al. (2024b)) for non-quantized autoregressive image generation, which utilizes non-quantized vectors as visual tokens and performs set-by-set autoregressive prediction. While both are non-quantized autoregressive approaches, it is non-trivial at all from MAR to **NOVA**: **1) NOVA** solves the challenges including efficiency, scalability, and mask schedule when learning more complex text-to-image generation instead of class-to-image generation. **2) NOVA** first predicts temporal frames sequentially and then predicts spatial sets within each frame. **NOVA** is the first to enable a non-quantized autoregressive model for video generation.

Specifically, **NOVA** predicts each frame in a casual order temporally, and predicts each token set in a random order spatially. In this way, text-to-video generation can be regarded as a fundamental task that implicitly and comprehensively encompasses various generation tasks (See Figure 1), including text-to-image, image-to-video, text&image-to-video, *etc*. With non-quantized tokenizers and a flexible autoregressive framework, **NOVA** simultaneously takes advantage of **1)** high-fidelity and compact visual compression for low cost in training and inference, and **2)** in-context abilities for integrating multiple visual generation tasks in a unified model.

For text-to-video generation, **NOVA** surpasses autoregressive counterparts in data efficiency, inference speed, and video fluency, while matching the performance of diffusion models of similar scale, *e.g.*, achieving a VBench (Huang et al. (2024)) score of 80.1 with a processing speed of 2.75 FPS[1], trained in only 342 GPU days on A100-40G. For text-to-image generation, NOVA achieves a GenEval (Ghosh et al. (2024)) score of 0.75, surpassing previous diffusion models with notably lower training cost, *e.g.*, only 127 GPU days for training this state-of-the-art 0.6B model. Additionally, **NOVA** also demonstrates strong zero-shot generalization across various contexts. We believe that **NOVA** paves the way for next-generation video generation, offering possibilities for real-time and infinite video generation, beyond Sora-like video diffusion models.

## 2 RELATED WORKS

### 2.1 DIFFUSION MODELS FOR VISUAL GENERATION

Diffusion models (Ho et al. (2020); Song et al. (2020)) have made significant advances in visual generation, including text-to-image tasks (Esser et al. (2024a); Betker et al. (2023a); Baldridge et al. (2024)) and text-to-video tasks (Brooks et al. (2024); Lin et al. (2024); Blattmann et al. (2023)). Image diffusion models typically model the joint distribution of fixed-length tokens in pixel (Ho et al. (2020); Nichol et al. (2021); Hoogeboom et al. (2023)) or latent space (Rombach et al. (2022a); Esser et al. (2024a); Betker et al. (2023a); Chen et al. (2023)). Besides, video diffusion models further introduce temporal layers to capture relationships between a fixed number of video frames. After training, additional tasks and modalities are added by incorporating extra inference tricks (Meng et al. (2021)), structure moderation (Blattmann et al. (2023); Esser et al. (2023); Liew et al. (2023)), and adapter layers (Zhang et al. (2023b); Guo et al. (2023)). Although these strategies can be composable, they stand in contrast to the autoregressive approaches (Kondratyuk et al. (2023); Hong et al. (2022); Radford (2018); Touvron et al. (2023)), which trains a single model end-to-end for multi-task learning,

---

[1]The 2.75 FPS is measured on a single NVIDIA A100-40G GPU using a batch size of 24.

offering notable context scalability and zero-shot generalizability across diverse application scenarios, especially in extending video generation duration.

## 2.2 AUTOREGRESSIVE MODELS FOR VISUAL GENERATION

**Raster-scan Autoregressive Models** are typically implemented on the discrete-valued RGB pixels (Kalchbrenner et al. (2017); Reed et al. (2017)) or latent space (Esser et al. (2021); Van Den Oord et al. (2017)), analogous to their language counterparts (Radford et al. (2019); Anil et al. (2023)). Recent studies involve autoregressive transformers to generate token sequences in the raster-scan order for image generation (Ramesh et al. (2021); Ding et al. (2021; 2022); Yu et al. (2022); Sun et al. (2024b)), and video generation (Yan et al. (2021); Kondratyuk et al. (2023); Nash et al. (2022); Wang et al. (2024b)). Specifically, VAR (Tian et al. (2024)) introduces next-scale prediction to progressively process the token-by-token sequence across multiple resolutions, leading to improved image quality.

**Masked Autoregressive Models** further develop a masked generative models (Chang et al. (2022)) to introduce a generalized autoregressive concept. They introduce a bidirectional transformer and predict randomly masked tokens by attending to unmasked conditions. This makes up for the suboptimal modeling and inefficient inference of sequentially line-by-line strategy, which inspires a series of subsequent works in text-to-image (Chang et al. (2023)) and text-to-video generation (Hong et al. (2022); Yu et al. (2023); Villegas et al. (2022)). Particularly, MAR (Li et al. (2024b)) decouples discrete tokenizers from autoregressive models and utilizes a diffusion procedure for per-token probability distributions. It is fully validated in the class-to-image field, holding great potential in the text-to-image domain. However, its application to text-to-video generation intuitively requires a masked autoregressive process across entire video frames, challenging multi-context learning and training efficiency. In contrast, our NOVA model breaks down video generation into frame-by-frame temporal predictions combined with spatial set-by-set predictions. This allows each frame to act as a meta causal unit, enabling extended video duration and zero-shot generalizability across various contexts. Besides, the subsequent spatial set-of-tokens prediction unlocks the power of bidirectional modeling patterns, enhancing inference efficiency while preserving visual quality and fidelity.

## 3 METHODOLOGY

We first review two categories of autoregressive video generation in Sec. 3.1. In Sec. 3.2-3.4, we introduce framework pipeline and implementation details of our NOVA, illustrated in Figure 2.

## 3.1 RETHINKING AUTOREGRESSIVE MODELS FOR VIDEO GENERATION

As mentioned above, we regard text-to-video generation and autoregressive (AR) model as the basic task and means, respectively. We briefly retrospect related technical background. There exist two types of AR video generation approaches: **(1) Token-by-token generation via raster-scan order.** These studies perform causal per-token prediction within video frame sequence (Kondratyuk et al. (2023)), and decode vision tokens sequentially following the raster scan ordering (Wang et al. (2024a)), which is defined as follows:

Table 1: Symbology Settings.

| | |
|---|---|
| $N, n$ | The number of all video tokens. |
| $F, f$ | The number of all video frames. |
| $K, k$ | The number of sets in an image. |

$$p\left(C, x_1, ..., x_N\right) = \prod_{n}^{N} p\left(x_n \mid C, x_1, ..., x_{n-1}\right), \tag{1}$$

where $C$ indicates various condition contexts, $e.g.$, label, text, image, and $etc$. Note that $x_n$ denotes $n$-th token of $N$ video raster-scale tokens. In contrast, **(2) Masked set-by-set generation in a random order** treats all tokens within each video frame equally, using a bidirectional transformer decoder for per-set prediction (Yu et al. (2023)). However, this generalized autoregressive (AR) model is trained using synchronous modeling on large, fixed-length video frames, which can lead to poor scalability in context and issues with coherence over longer video durations. Hence, NOVA proposes a novel solution by decoupling per-set generation within a single video frame from the per-frame prediction across the entire video sequence. This allows NOVA to better handle both temporal causality and spatial relationships, providing a more flexible and scalable AR framework.

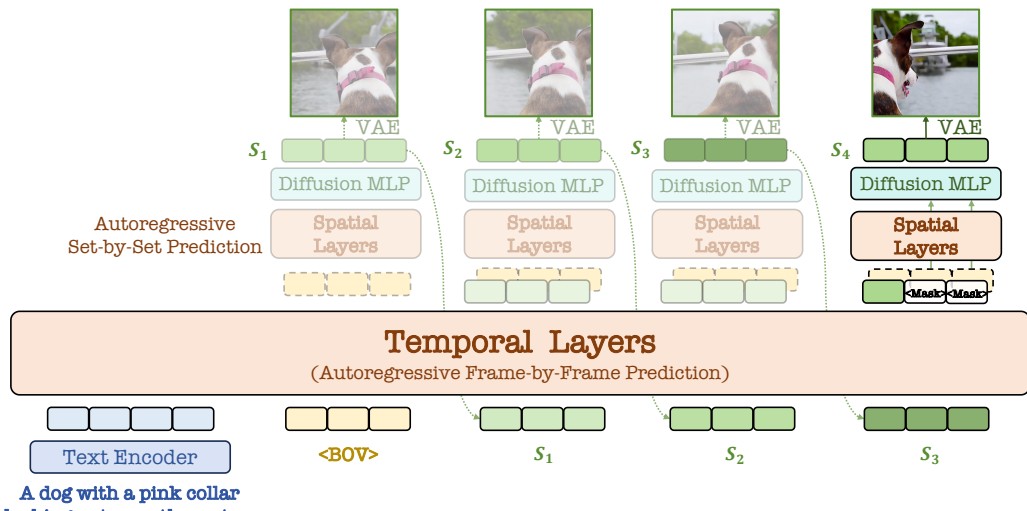

Figure 2: **NOVA framework and the inference process.** With text inputs, NOVA performs autoregressive generation via temporal frame-by-frame prediction and spatial set-by-set prediction. Finally, we implement diffusion denoising in a continuous-values space.

## 3.2 TEMPORAL AUTOREGRESSIVE MODELING VIA FRAME-BY-FRAME PREDICTION

Inspired by (Zhuo et al. (2024)), we use a pre-trained language model (Javaheripi et al. (2023)) to encode text prompts to features. To better control video dynamics, we use OpenCV (cv2) (Bradski (2000)) to compute the optical flow of sampled video frames. The average flow magnitude is used as a motion score and integrated with the prompt. Besides, we employ open-source 3D variational autoencoder (VAE) (Lin et al. (2024)) with a temporal stride of 4 and a spatial stride of 8 to encode the video frames to the latent space. We add an additional learnable patch embedding layer with a spatial stride of 4 to align channels of latent video to the subsequent transformer. Notably, next-token prediction in early AR models seems counter-intuitive for undirected visual patches within a single image and suffers from high latency during inference. In contrast, video frames can naturally be viewed as a causal sequence, with each frame acting as a meta unit for AR generation. Therefore, we implement block-wise causal masking attention depicted in Figure 3(a), ensuring that each frame can only attend to the text prompts, video flow, and its preceding frames, while allowing all current frame tokens to be visible to each other as follows:

$$p\left(P, m, B, S_1, ..., S_F\right) = \prod_f^F p\left(S_f \mid P, m, B, S_1, ..., S_{f-1}\right), \tag{2}$$

where $P, m$ indicate text prompts and video flow respectively. Here, $S_f$ denotes the overall tokens of $f$-th video frame, and $B$ represent learnable begin-of-video (BOV) embeddings for predicting the initial video frame, the number of which corresponds to the patch number of one single frame. Note that we add 1-D and 2-D sine-cosine embeddings (Vaswani et al. (2017)) with video frame features to indicate time and position information respectively, which are convenient for temporal and spatial extrapolation. From equation 2, we can reformulate text-to-image and image-to-video generation as $p\left(S_1 \mid P, m, B\right)$ and $p\left(S_f \mid \varnothing, m, B, S_1, ..., S_{f-1}\right)$. This generalized causal process can synchronously model the condition contexts for each video frame, greatly enhancing training efficiency, and allowing the kv-cache technology for fast decoding procedure during inference.

## 3.3 SPATIAL AUTOREGRESSIVE MODELING VIA SET-BY-SET PREDICTION

Inspired by (Chang et al. (2022); Li et al. (2024b)), we define each token set with multiple tokens from random directions as a meta causal token in Figure 3(b), facilitating a generalized AR process with efficient parallel decoding. Notably, we tried to utilize the temporal layers' outputs targeting one frame as indicator features to assist the spatial layers, gradually decoding all randomly masked token sets within the corresponding image. However, this approach resulted in image structure collapse and

**(a) Temporal Per-token vs. Per-frame Prediction**    **(b) Spatial Per-token vs. Per-set Prediction**

Figure 3: Overview of our block-wise temporal and spatial generalized autoregressive attention. Different from per-token generation, NOVA regressively predicts each frame in a casual order across the temporal scale, and predicts each token set in a random order across the spatial scale.

inconsistent video fluency over the increasing number of frames. We hypothesize that this occurs because the indicator features from adjacent frames are similar, making it difficult to accurately learn continuous and imperceptible motion changes without explicit modeling. Besides, the indicator features derived from the ground-truth contextual frame during training contribute to weak robustness and stability of spatial AR layers against cumulative inference errors.

To address this issue, we introduce a Scaling and Shift Layer that reformulates cross-frame motion changes by learning relative distribution variations within a unified space, rather than directly modeling the unreferenced distribution of the current frame. Notably, we select the BOV-attended output of the temporal layers as the anchor feature set, as it serves as the initial feature set with significantly less noise accumulation than subsequent frame feature sets. Specifically, we first translate the features from current frame set into dimension-wise variance and mean parameters $\gamma$ and $\beta$ via multi-layer perception (MLP). After that, we affine the normalized features from the anchor set into indicator features $S'_f$ via channel-wise scale and shift operation. Specially, we explicitly set $\gamma = 1$ and $\beta = 0$ for the first frame. With unmasked token features, we predict randomly masked visual tokens in a set-by-set order through a bidirectional paradigm, which can be formulated as follows:

$$p\left(S'_f, S_{(f,1)}, ..., S_{(f,K)}\right) = \prod_{k}^{K} p\left(S_{(f,k)} \mid S'_f, S_{(f,1)}, ..., S_{(f,k-1)}\right), \qquad (3)$$

where $S'_f$ denotes the indicator features for generating $f$-th video frame, and $S_{(f,k)}$ denotes $k$-th token set of $f$-th video frame. We add 2-D sine-cosine embeddings with masked and unmasked tokens to indicate their relative position. This generalized spatial AR prediction leverages powerful bidirectional patterns within single-image tokens and achieves efficient inference with parallel masked decoding. *Notably, we incorporate post-norm layers before the residual connections in both temporal and spatial AR layers.* Our empirical findings show that this design effectively addresses architectural and optimization challenges that previously hindered stable training in generalized video generation.

### 3.4 DIFFUSION PROCEDURE DENOISING FOR PER-TOKEN PREDICTION

During training, we import *diffusion loss* (Li et al. (2024b)) to estimate per-token probability in a continuous-valued space. For example, we define one ground-truth token as $x_n$ and corresponding NOVA's output as $z_n$. The loss function can be formulated as a denoising criterion:

$$\mathcal{L}(x_n \mid z_n) = \mathbb{E}_{\varepsilon,t}\left[\left\|\epsilon - \epsilon_\theta\left(x_n^t \mid t, z_n\right)\right\|^2\right]. \qquad (4)$$

Here $\epsilon$ is a Gaussian vector sampled from $\mathcal{N}(\mathbf{0}, \mathbf{I})$, and noisy data $x_n^t = \sqrt{\bar{\alpha}_t}x_n + \sqrt{1 - \bar{\alpha}_t}\epsilon$, where $\bar{\alpha}_t$ is a noise schedule (Nichol & Dhariwal (2021)) indexed by a time step $t$. The noise estimator $\epsilon_\theta$ is multiple MLP blocks parameterized by $\theta$. The notation $\epsilon_\theta(x_n^t \mid t, z_n)$ means that this network takes $x_n^t$ as the input, and is conditional on both $t$ and $z_n$. We follow (Li et al. (2024b)) to sample $t$ by 4 times during training for each image.

During inference, we sample $x_n^T$ from a random Gaussian noise $\mathcal{N}(\mathbf{0}, \mathbf{I})$ and denoise it step-by-step by sequentially sampling $x_n^T$ to $x_n^0$ via $x_n^{t-1} = \frac{1}{\sqrt{\alpha_t}}\left(x_n^t - \frac{1-\alpha_t}{\sqrt{1-\bar{\alpha}_t}}\epsilon_\theta\left(x_n^t \mid t, z_n\right)\right) + \sigma_t\epsilon$, where $\sigma_t$ is the noise level at time step $t$, and $\epsilon$ is sampled from the Gaussian distribution $\mathcal{N}(\mathbf{0}, \mathbf{I})$.

## 4 EXPERIMENT

### 4.1 EXPERIMENT SETUP

**Datasets.** We involve several diverse, curated, and high-quality datasets to facilitate the training of our NOVA. For text-to-image training, we initially curate 16M image-text pairs sourced from DataComp (Gadre et al. (2024)), COYO (Byeon et al. (2022)), Unsplash (UnsplashTeam (2020)), and JourneyDB (Sun et al. (2024a)). To explore the scaling properties of NOVA, we expanded the dataset to approximately 600M image-text pairs by selecting more images that have a minimum aesthetic score of 5.0 from LAION (Schuhmann et al. (2022)), DataComp and COYO. For text-to-video training, we select 19M video-text pairs on a subset (Lin et al. (2024)) of Panda-70M (Chen et al. (2024b)) and internal video-text pairs. We further collect 1M of high-resolution video-text pairs from Pexels (PexelsTeam (2014)) to fine-tune our final video generation model. Following (Diao et al. (2024)), we train a caption engine based on Emu2-17B (Sun et al. (2023)) model to create high-quality descriptions for our image and video datasets. The maximum text length is set to 256.

**Architectures.** We mostly follow (Li et al. (2024b)) to build NOVA's spatial AR layer and denoising MLP block, including a layer sequence of LayerNorm (Lei Ba et al. (2016)), AdaLN (Huang & Belongie (2017)), linear layer, SiLU activation (Elfwing et al. (2018)), and another linear layer. We configure the temporal encoder, spatial encoder, and decoder with 16 layers each, using a dimension of 768 (0.3B), 1024 (0.6B) or 1536 (1.4B). The denoising MLP consists of 3 blocks with a dimension of 1280. The spatial layers adopt the encoder-decoder architecture of MAR (Li et al. (2024b)), similar to MAE (He et al. (2022)). Specifically, the encoder processes the visible patches for reconstruction. The decoder further processes visible and masked patches for generation. To capture the image latent features, we employ a pre-trained and frozen VAE from (Lin et al. (2024)), which achieves $4\times$ compression in the temporal dimension and $8 \times 8$ compression in the spatial dimension. We adopt the masking and diffusion schedulers from (Li et al. (2024b); Nichol & Dhariwal (2021)), using a masking ratio between 0.7 and 1.0 during training, and progressively reducing it from 1.0 to 0 following a cosine schedule (Chang et al. (2023)) during inference. In line with common practice (Ho et al. (2020)), we train with a 1000-step noise schedule but default to 100 steps for inference.

**Training details.** NOVA is trained with sixteen A100 (40G) nodes. We utilize the AdamW optimizer (Loshchilov et al. (2017)) ($\beta_1 = 0.9, \beta_2 = 0.95$) with a weight decay of 0.02 and a base learning rate of 1e-4 in all experiments. The peak learning rate is adjusted for different batch sizes during training using the scaling rule (Goyal (2017)) : lr = base_lr $\times$ batchsize$/256$. We train text-to-image models from scratch and then load these weights to train text-to-video models.

**Evaluation.** We use T2I-CompBench (Huang et al. (2023)), GenEval (Ghosh et al. (2024)) and DPG-Bench (Hu et al. (2024)) to assess the alignment between the generated images and text condition. We generate image samples for each of the original or rewritten (Wang et al. (2024a)) text prompts. Each image sample has a resolution of 512×512 or 1024×1024. We use VBench (Huang et al. (2024)) to evaluate the capacity of text-to-video generation across 16 dimensions. For a given text prompt, we randomly generate 5 samples, each with a video size of 33×768×480. We employ classifier-free guidance (Ho & Salimans (2022)) with a value of 7.0 along with 128 autoregressive steps to enhance the quality of the generated images and videos in all evaluation experiments.

### 4.2 MAIN RESULTS

**NOVA outperforms existing text-to-image models with superior performance and efficiency.** In Table 2, we compare NOVA with several recent text-to-image models, including PixArt-$\alpha$ (Chen et al. (2023)), SD v1/v2 (Rombach et al. (2022b)), SDXL (Podell et al. (2023)), DALL-E2 (Ramesh et al. (2022)), DALL-E3 (Betker et al. (2023b)), SD3 (Esser et al. (2024b)), LlamaGen (Sun et al. (2024b) and Emu3 (Wang et al. (2024a)). After text-to-image training, NOVA achieves state-of-the-art performance on the GenEval benchmark, especially in generating a specified number of targets. Notably, NOVA also achieves leading results on T2I-CompBench and DPG-Bench, excelling at both the small model scale and data scale (requiring only 16% training overhead of the best competitor PixArt-$\alpha$). *Last but not least, our text-to-video model outperforms most specialized text-to-image models*, *e.g*., SD v1/v2, SDXL and DALL-E2. This underscores the robustness and versatility of our model in multi-context scenarios, with text-to-video generation as the fundamental training task.

Table 2: **Text-to-image evaluation on various benchmarks.** The best and second-best results are in blue and green . The data is from Huang et al. (2023),Wang et al. (2024a) and Esser et al. (2024b).

| Model | ModelSpec | | T2I-CompBench | | | GenEval | | | | | | | DPG-Bench | A100 days |
|---|---|---|---|---|---|---|---|---|---|---|---|---|---|---|
| | #params | #images | Color | Shape | Texture | Overall | Single | Two | Counting | Colors | Position | ColorAttr | Overall | |
| *Diffusion models* | | | | | | | | | | | | | | |
| PixArt-α | 0.6B | 25M | 68.86 | 55.82 | 70.44 | 0.48 | 0.98 | 0.50 | 0.44 | 0.80 | 0.08 | 0.07 | 71.11 | 753 |
| SD v1.5 | 1B | 2B | 37.50 | 37.24 | 42.19 | 0.43 | 0.97 | 0.38 | 0.35 | 0.76 | 0.04 | 0.06 | 63.18 | - |
| SD v2.1 | 1B | 2B | 56.94 | 44.95 | 49.82 | 0.50 | 0.98 | 0.37 | 0.44 | 0.85 | 0.07 | 0.17 | - | - |
| SDXL | 2.6B | - | 63.69 | 54.08 | 56.37 | 0.55 | 0.98 | 0.44 | 0.39 | 0.85 | 0.15 | 0.23 | 74.65 | - |
| DALL-E2 | 6.5B | 650M | 57.50 | 54.64 | 63.74 | 0.52 | 0.94 | 0.66 | 0.49 | 0.77 | 0.10 | 0.19 | - | - |
| DALL-E3 | - | - | 81.10 | 67.50 | 80.70 | 0.67 | 0.96 | 0.87 | 0.47 | 0.83 | 0.43 | 0.45 | 83.50 | - |
| SD3 | 2B | - | - | - | - | 0.62 | 0.98 | 0.74 | 0.63 | 0.67 | 0.34 | 0.36 | 84.10 | - |
| *Autoregressive models* | | | | | | | | | | | | | | |
| LlamaGen | 0.8B | 60M | - | - | - | 0.32 | 0.71 | 0.34 | 0.21 | 0.58 | 0.07 | 0.04 | - | - |
| Emu3 (+ Rewriter) | 8B | - | 79.13 | 58.46 | 74.22 | 0.66 | 0.99 | 0.81 | 0.42 | 0.80 | 0.49 | 0.45 | 81.60 | - |
| NOVA (512×512) | 0.6B | 16M | 70.75 | 55.98 | 69.79 | 0.66 | 0.98 | 0.85 | 0.58 | 0.83 | 0.20 | 0.48 | 81.76 | 127 |
| + Rewriter | 0.6B | 16M | 83.02 | 61.47 | 75.80 | 0.75 | 0.98 | 0.88 | 0.62 | 0.82 | 0.62 | 0.58 | - | 127 |
| + Videos | 0.6B | 36M | 71.80 | 47.86 | 65.31 | 0.55 | 0.98 | 0.56 | 0.48 | 0.75 | 0.15 | 0.41 | 81.77 | 342 |
| + Videos & Rewriter | 0.6B | 36M | 81.36 | 59.16 | 72.45 | 0.71 | 0.98 | 0.83 | 0.52 | 0.81 | 0.58 | 0.51 | - | 342 |
| NOVA (1024×1024) | 0.3B | 600M | 73.35 | 57.28 | 70.09 | 0.67 | 0.98 | 0.86 | 0.53 | 0.84 | 0.32 | 0.52 | 80.60 | 267 |
| NOVA (1024×1024) | 0.6B | 600M | 74.72 | 56.99 | 69.50 | 0.69 | 0.98 | 0.89 | 0.56 | 0.84 | 0.32 | 0.56 | 82.25 | 320 |
| NOVA (1024×1024) | 1.4B | 600M | 74.30 | 57.14 | 70.00 | 0.71 | 0.99 | 0.91 | 0.62 | 0.85 | 0.33 | 0.56 | 83.01 | 608 |

Table 3: **Text-to-video evaluation on VBench.** We have classified existing video generation methods into different categories for better clarity. The baseline data is sourced from Huang et al. (2024).

| Model | #params | #videos | latency | Total Score | Quality Score | Semantic Score | Aesthetic Quality | Object Class | Multiple Objects | Human Action | Spatial Relationship | Scene |
|---|---|---|---|---|---|---|---|---|---|---|---|---|
| *Closed-source models* | | | | | | | | | | | | |
| Gen-2 | - | - | - | 80.58 | 82.47 | 73.03 | 66.96 | 90.92 | 55.47 | 89.20 | 66.91 | 48.91 |
| Kling (2024-07) | - | - | - | 81.85 | 83.39 | 75.68 | 61.21 | 87.24 | 68.05 | 93.40 | 73.03 | 50.86 |
| Gen-3 | - | - | - | 82.32 | 84.11 | 75.17 | 63.34 | 87.81 | 53.64 | 96.4 | 65.09 | 54.57 |
| *Diffusion models (w/ SD init)* | | | | | | | | | | | | |
| LaVie | 3B | 25M | - | 77.08 | 78.78 | 70.31 | 54.94 | 91.82 | 33.32 | 96.8 | 34.09 | 52.69 |
| Show-1 | 4B | 10M | - | 78.93 | 80.42 | 72.98 | 57.35 | 93.07 | 45.47 | 95.60 | 53.50 | 47.03 |
| AnimateDiff-v2 | 1B | 10M | - | 80.27 | 82.90 | 69.75 | 67.16 | 90.90 | 36.88 | 92.60 | 34.60 | 50.19 |
| VideoCrafter-v2.0 | 2B | 10M | - | 80.44 | 82.20 | 73.42 | 63.13 | 92.55 | 40.66 | 95.00 | 35.86 | 55.29 |
| T2V-Turbo (VC2) | 2B | 10M | - | 81.01 | 82.57 | 74.76 | 63.04 | 93.96 | 54.65 | 95.20 | 38.67 | 55.58 |
| *Diffusion models* | | | | | | | | | | | | |
| OpenSora-v1.1 | 1B | 10M | 48s | 75.66 | 77.74 | 67.36 | 50.12 | 86.76 | 40.97 | 84.20 | 52.47 | 38.63 |
| OpenSoraPlan-v1.1 | 1B | 4.5M | 60s | 78.00 | 80.91 | 66.38 | 56.85 | 76.30 | 40.35 | 86.80 | 53.11 | 27.17 |
| OpenSora-v1.2 | 1B | 32M | 55s | 79.76 | 81.35 | 73.39 | 56.85 | 82.22 | 51.83 | 91.20 | 68.56 | 42.44 |
| CogVideoX | 2B | 35M | 90s | 80.91 | 82.18 | 75.83 | 60.82 | 83.37 | 62.63 | 98.00 | 69.90 | 51.14 |
| *Autoregressive models* | | | | | | | | | | | | |
| CogVideo | 9B | 5.4M | - | 67.01 | 72.06 | 46.83 | 38.18 | 73.4 | 18.11 | 78.20 | 18.24 | 28.24 |
| Emu3 | 8B | - | - | 80.96 | 84.09 | 68.43 | 59.64 | 86.17 | 44.64 | 77.71 | 68.73 | 37.11 |
| NOVA | 0.6B | 20M | 12s | 78.48 | 78.96 | 76.57 | 54.52 | 91.36 | 73.46 | 91.20 | 66.37 | 50.16 |
| + Rewriter | 0.6B | 20M | 12s | 80.12 | 80.39 | 79.05 | 59.42 | 92.00 | 77.52 | 95.20 | 77.52 | 54.06 |

**NOVA rivals diffusion text-to-video models and significantly suppresses the AR counterpart.** We emphasize that the current version of our NOVA is designed to generate videos at 33 frames and can extend video length through the *pre-filling* of recently generated frames. We perform a quantitative analysis comparing NOVA against open-source and proprietary text-to-video models. As shown in Table 3, despite its significantly smaller size (0.6B vs. 9B), NOVA remarkably outperforms CogVideo (Hong et al. (2022)) across a variety of text-to-video evaluation metrics. It also matches the latest SOTA model Emu3's (Wang et al. (2024a)) performance (80.12 vs. 80.96) with a significantly smaller size (0.6B vs. 8B). Additionally, we compared NOVA with state-of-the-art diffusion models. This includes both the closed-source models (Runway (2023); Kuaishou (2024); Runway (2024)), as well as the open-source alternatives (Wang et al. (2023); Zhang et al. (2023a); Guo et al. (2024); Chen et al. (2024a); Li et al. (2024a); Zheng et al. (2024); Lin et al. (2024); Yang et al. (2024)).

## 4.3 QUALITATIVE RESULTS

**High-fidelity image and high-fluency video.** We present a qualitative comparison of current leading image generation methods in Figure 4. NOVA demonstrates strong visual quality and fidelity across a range of prompts. We present text-to-video visualizations in Figure 5, which highlight NOVA's ability to capture multi-view perspectives, smooth object motion, and stable scene transitions.

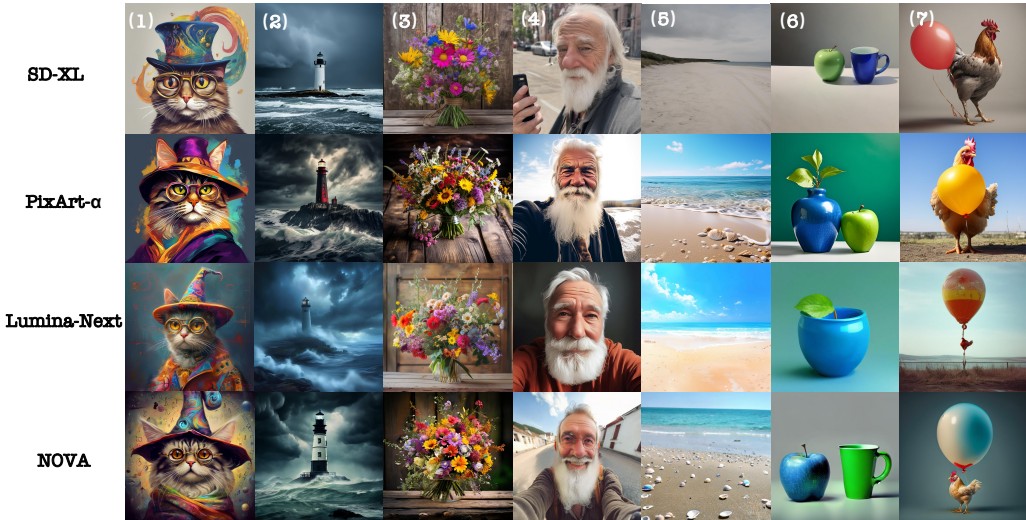

Figure 4: **Text-to-image generation.** Text prompts from left to right: (1) "A digital artwork of a cat styled in a whimsical fashion...", (2) "A solitary lighthouse standing tall against a backdrop of stormy seas and dark, rolling clouds", (3) "A vibrant bouquet of wildflowers on a rustic wooden table", (4) "A selfie of an old man with a white beard", (5) "A serene, expansive beach with no people", (6) "A blue apple and a green cup." and (7) "A chicken on the bottom of a balloon."

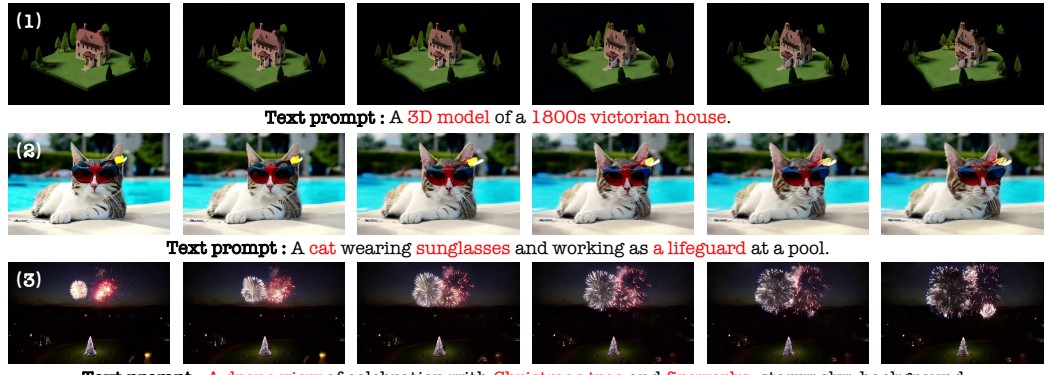

Figure 5: **Text-to-video generation.** We highlight the keywords in red color. NOVA follows the text prompts and vividly captures the motion of subjects (i.e., 3D model, cat and fireworks).

**Zero-shot generalization on video extrapolation.** By pre-filling generated frames, NOVA can produce videos that surpass the training length. For example, by shifting both the text and BOV embeddings, we generate videos that are up to twice the original length, as shown in Figure 6.

**Zero-shot generalization on multiple contexts.** By pre-filling the reference image, NOVA can generate videos from images, either with or without accompanying text. In Figure 7, we provide a qualitative example. We show that NOVA can simulate realistic motions without text prompts.

## 4.4 ABLATION STUDY

**Effectiveness of temporal autoregressive modeling.** To highlight the advantages of temporal AR, we have facilitated spatial AR to finish video generation task. We observe less subject movement in videos under the same training iterations (Figure 8). Additionally, in zero-shot generalization across various contexts or video extrapolation, the network output exhibited more artifacts and temporal inconsistencies. Furthermore, this approach is not compatible with kv-cache acceleration during inference, leading to a linear increase in latency with the number of video frames.

**Effectiveness of Scaling and Shift Layer.** To capture cross-frame motion changes, we employ a simple yet effective scaling and shifting layer to explicitly model the relative distribution from the BOV-attended feature space. In Figure 9(a), we demonstrate that this approach significantly reduces

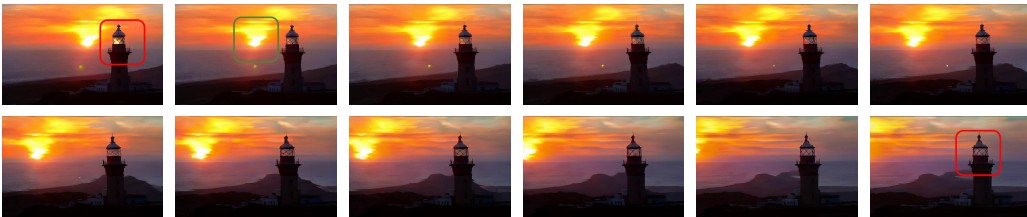

A lighthouse ... a vibrant sunset sky... to a warm palette of oranges and yellows ... has a classic design with a domed top and a lantern room where the light would be housed, ...to intensify.

Figure 6: **Zero-shot video extrapolation.** We highlight the subjects in red and green respectively. The top images are generated, while the bottom images are extrapolated.

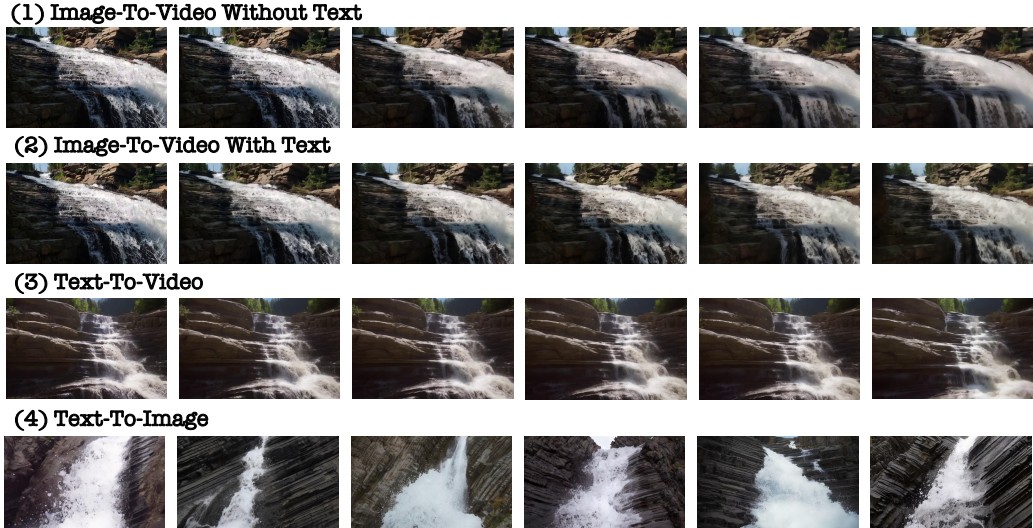

**(1) Image-To-Video Without Text**

**(2) Image-To-Video With Text**

**(3) Text-To-Video**

**(4) Text-To-Image**

**Text :** A cascade of water rushes down a rocky incline, frothing and churning as it descends, is surrounded by rugged, layered rock formations.

Figure 7: **Zero-shot generalization on multiple contexts.** It is evident that NOVA successfully maintains temporal consistency in objects, both with and without text. Such as ensuring "water continues to flow smoothly." This highlights NOVA's capability for zero-shot multitasking.

the drift between text-to-image and image-to-video generation losses. As we gradually decrease the inner rank of the MLP, the training difficulty increases, leading to a more comprehensive and robust learning process for the network. However, extremely low rank values pose challenges for motion modeling, as they significantly limit the layer's representation capability (Figure 10). The rank is set to 24 by default in all text-to-video experiments, resulting in more accurate motion predictions.

**Effectiveness of Post-Norm Layer.** Training large-scale image and video generation models (Ding et al. (2021); ChameleonTeam (2024)) from scratch often poses significant challenges with mixed precision, which is also observed in other visual recognition methods (Liu et al. (2022)). As shown in Figure 9(b), the training process with pre-normalization (Dosovitskiy et al. (2021)) suffers from numerical overflow and variance instability. We attempted various regularization techniques on the residual branch, such as stochastic depth (Huang et al. (2016)) and residual dropout (Vaswani et al. (2017)), but found them to be less effective. Inspired by (Liu et al. (2022)), we introduce post-normalization and empirically discover that it can effectively mitigate the residual accumulation of output embeddings compared to pre-normalization, resulting in a more stable training process.

## 5 CONCLUSION

In this paper, we present **NOVA**, a novel autoregressive model designed for both text-to-image and text-to-video generation. **NOVA** delivers exceptional image quality and video fluency while significantly minimizing training and inference overhead. Our key designs include temporal frame-by-frame prediction, spatial set-by-set generation, and continuous-space autoregressive modeling across various contexts. Extensive experiments demonstrate that **NOVA** achieves near-commercial

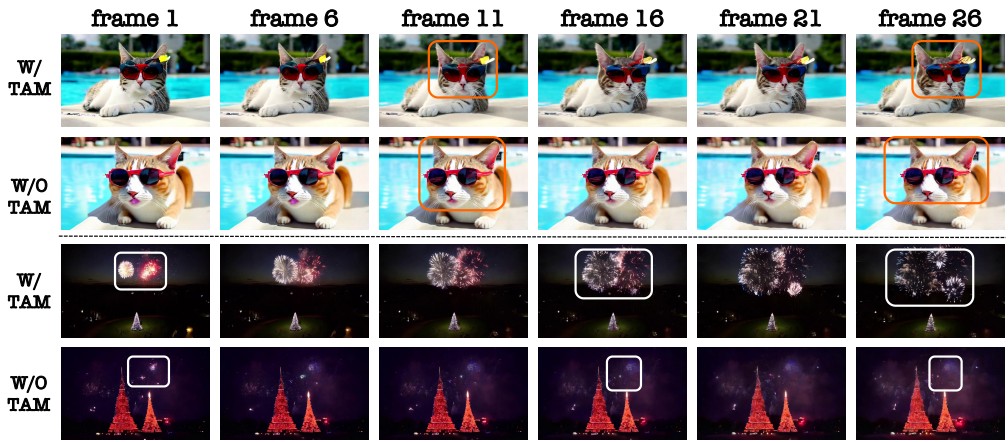

Figure 8: **Temporal autoregressive modeling (TAM) for video generation.** We highlight the subtle changes in frames generated from the same prompt. Compared to spatial-only autoregressive method, the inclusion of TAM enables NOVA to more accurately capture the dynamics of subject movement.

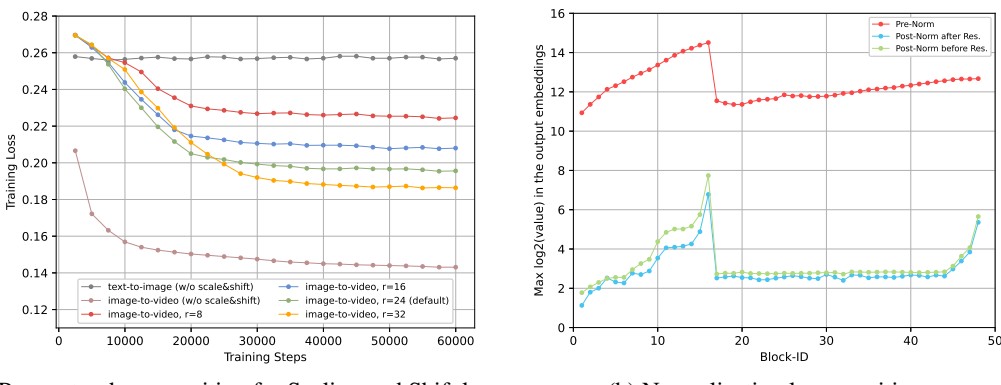

(a) Parameter decomposition for Scaling and Shift layer.  (b) Normalization layer position.

Figure 9: **Ablation studies on NOVA's architecture components.** We carefully examine the two key stability factors in large-scale video generation training, as illustrated in (a) and (b).

quality in image generation, alongside promising fidelity and fluency in video generation. **NOVA** paves the way for next-generation video generation and world models. It offers valuable insights and possibilities for real-time and infinite video generation, going beyond Sora-like video diffusion models. As a first step, we will continue scalable experiments with larger models and data scaling to explore NOVA's limits in future work.

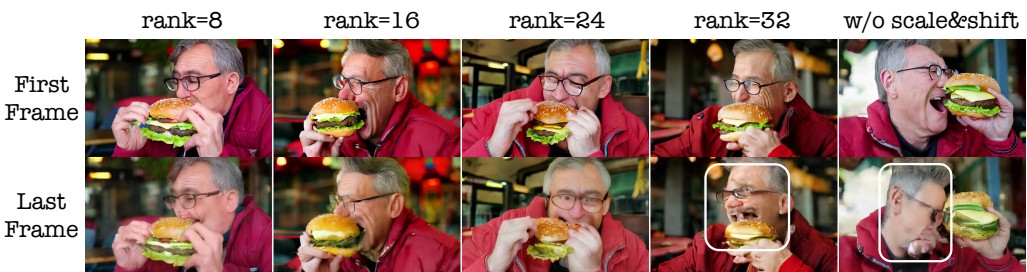

Figure 10: **Visualization of decomposition ranks in the Scaling and Shift layer.** The first row displays the results of the first frame, while the second row presents the results of the last frame.

**Acknowledgement.** We thank Fan Zhang for the valuable advice on Figure 2 of this work, and Xiaotong Li, Jinming Wu, Chengyue Wu, Zhen Li, Quan Sun, Yueze Wang, Jinsheng Wang and Xiaosong Zhang for the insightful discussions. We also acknowledge the support provided by other colleagues at the Beijing Academy of Artificial Intelligence (BAAI) towards this project. This work was supported by the National Key R&D Program of China (2022ZD0116302), in part by Hainan Provincial Natural Science Foundation of China under No.624LALH008, and in part by the Program for Youth Innovative Research Team of BUPT under No. 2023YQTD02.

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
