# Autoregressive Video Generation without Vector Quantization

## Appendix

We strictly publish our code and pretrained models to improve interpretability and assure reproducibility. Here, more implementation details and ablation experiments are organized as follows:

- Architecture details of Scaling and Shift layer (Sec. A)
- Normalization configurations (Sec. B)
- Video extrapolation evaluations (Sec. C)
- Inference time analysis (Sec. D)
- Ablations on the impact of temporal autoregressive modeling (Sec. E)
- Comprehensive DPG-Bench evaluation results (Sec. F)
- More text-to-image visualizations (Sec. G)
- More text-to-video visualizations (Sec. H)

## A   Architecture details of Scaling and Shift layer

The Scaling and Shift Layer is implemented as an adaptive normalization layer, adopting the design initially proposed by FiLM (Perez et al. (2018)) and AdaIN (Huang & Belongie (2017)). While many previous methods have primarily utilized adaptive normalization for controllable image generation, such as in StyleGAN (Karras et al. (2019)), or for conditional modeling within Diffusion Transformers, like DiT (Peebles & Xie (2023)), NOVA innovatively applies this technique to manage the cumulative inference errors in autoregressive video generation. We employ a two-layer MLP to optimize low-rank decomposition for motion changes, as shown in the Figure 1. Specifically, we refer `AdaLayerNorm` and decompose the motion changes into mean and variance parameters, which are further used to apply the affine transformation on BOV embeddings.

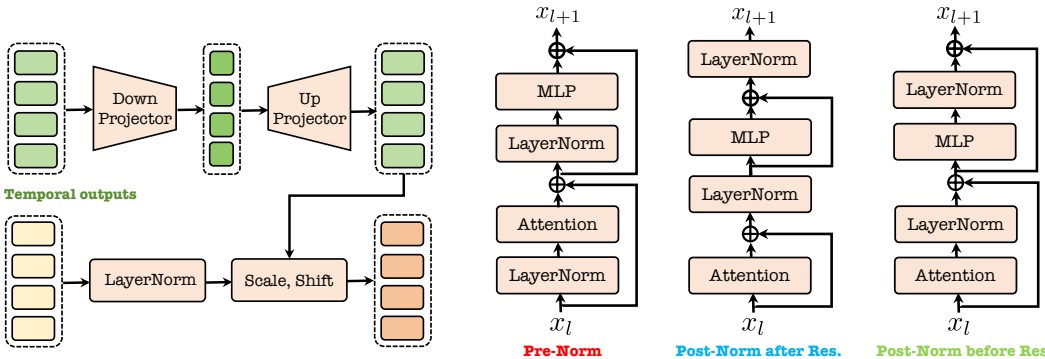

Figure 1: **Scaling and Shift layer.** We reformulate cross-frame motion changes by learning relative distribution variations within a unified space based on BOV tokens, rather than directly modeling the unreferenced distribution of the current frame.

Figure 2: **Three normalization architectures.** We summarize various configurations including the pre-normalization layer (left), the post-normalization layer after residual addition (middle), and the post-normalization layer before residual addition (right). Here Post-Norm before Res is our standard design.

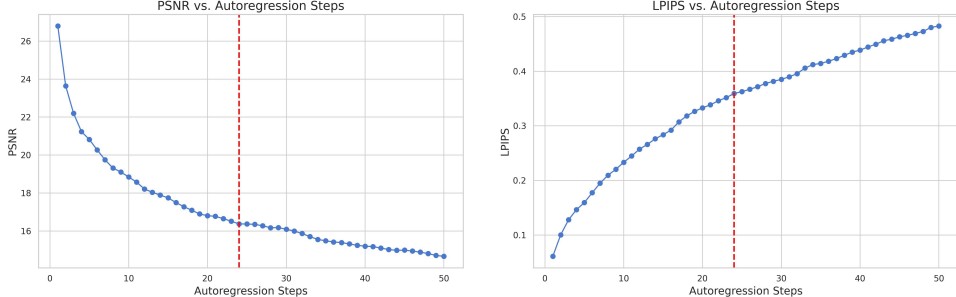

Figure 3: **PSNR and LPIPS metrics over 50 autoregressive steps in video extrapolation.** Due to the $4\times$ downsampling rate of VAE in temporal scale, each autoregressive step generates four frames. The vertical red line marks the point where the extrapolation reaches $3\times$ training length.

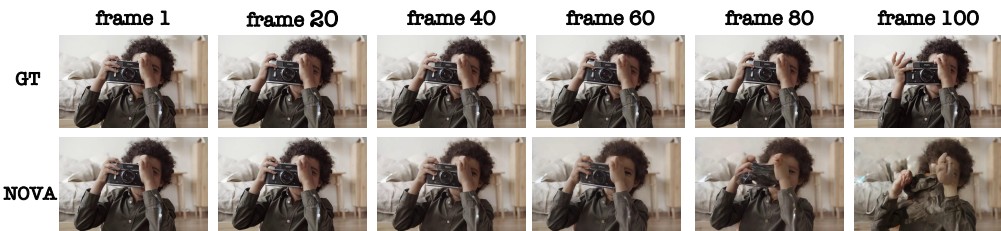

Figure 4: **Visualization of video extrapolation.** Although the metrics indicate a decline, the generated frames still closely resemble the original video in content and overall image quality. Visualization suggests that the model can extrapolate up to $3\times$ training length.

## B  NORMALIZATION CONFIGURATIONS

NOVA employs an improved normalization configuration that can effectively control the numerical boundaries of the output embeddings of each Transformer block while also maintaining the identity transformation of residual connections. We illustrate the three common normalization configurations in Figure 2, and NOVA uses the *post-normalization before residual addition* by default.

## C  VIDEO EXTRAPOLATION EVALUATIONS

Video extrapolation represents a significant challenge, being an out-of-domain generalization issue. To assess our model's performance, we curated a test set comprising 200 videos. For each video, the task involved generating subsequent frames from the initial frame and a textual prompt, effectively converting an image and text into a video sequence. We utilized LPIPS (Zhang et al. (2018)) and PSNR metrics to evaluate the video extrapolation capabilities of our model.

During the extrapolation process, it was observed that the generated frames started to deviate from the ground truth after a few iterations. This is mainly due to the difficulty in accurately capturing video dynamics, causing minor discrepancies to accumulate. As a result, per-frame PSNR values decrease, while LPIPS scores increase over time (Figure 3). Nevertheless, the generated frames exhibit a high degree of similarity to the original video in terms of both content and image quality in Figure 4. This highlights the robustness of our temporal autoregressive approach in video extrapolation.

## D  INFERENCE TIME ANALYSIS

We report inference times on a single NVIDIA A100 GPU (40GB) with a batch size of 24 in Table 1. In each video, the temporal layers require only 0.03 seconds, compared to 11.97 seconds for the

spatial layers, highlighting the exceptional efficiency of the temporal layers. While NOVA is already efficient in text-to-video generation, there is potential for further acceleration in the spatial layers.

Table 1: **Inference time analysis for different layers.**

| Resolution | Temporal Layers Time | Spatial Layers Time | Total Time |
|---|---|---|---|
| 29×768×480 | 0.03s | 11.97s | 12s |

# E    ABLATIONS ON THE IMPACT OF TEMPORAL AUTOREGRESSIVE MODELING

Under the same experimental settings, we evaluated VBench results both with and without using TAM (Temporal Autoregressive Modeling) to highlight its significance. Our findings are summarized as follows: **(1) Efficient Motion Modeling:** We observed that the total score was marginally lower without TAM compared to NOVA (75.38 vs. 75.84), especially in the dynamic degree metric, which showed a more pronounced decline (11.38 vs. 23.27). We hypothesize that while bidirectional attention enhances model capacity, it requires more extensive data and longer training times to capture subtle motion changes compared to causal models. **(2) Efficient Video Inference:** Thanks to the kv-cache technology and frame-by-frame autoregressive processing, NOVA's inference time is much faster compared to methods without TAM, with a greater speed advantage for longer videos.

Table 2: **Performance comparison on temporal autoregressive modeling.**

| Model | Total Score | Dynamic Degree | Infer Time |
|---|---|---|---|
| NOVA | 75.84 | 23.27 | 12s |
| NOVA (w/o TAM) | 75.38 | 11.38 | 39s |

# F    COMPREHENSIVE DPG-BENCH EVALUATION RESULTS

We provide detailed DPG-Bench scores in Table 3. While NOVA outperforms most models of comparable size and matches the overall score of state-of-the-art models, we observe that increasing the model scale results in marginal improvements and does not boost the text rendering performance. This limitation may be attributed to our reliance on extensive web datasets, such as LAION and DataComp. In future work, we plan to focus on improving the quality of text-to-image data.

Table 3: **Comparison with state-of-the-art models on DPG-Bench.**

| Model | Overall | Global | Entity | Attribute | Relation | Other |
|---|---|---|---|---|---|---|
| *Diffusion models* | | | | | | |
| SD v1.5 (Rombach et al. (2022)) | 63.18 | 74.63 | 74.23 | 75.39 | 73.49 | 67.81 |
| PixArt-$\alpha$ (Chen et al. (2023)) | 71.11 | 74.97 | 79.32 | 78.60 | 82.57 | 76.96 |
| PixArt-$\sigma$ (Chen et al. (2024)) | 80.54 | 86.89 | 82.89 | 88.94 | 86.59 | 87.68 |
| Lumina-Next (Zhuo et al. (2024)) | 74.63 | 82.82 | 88.65 | 86.44 | 80.53 | 81.82 |
| SDXL (Podell et al. (2023)) | 74.65 | 83.27 | 82.43 | 80.91 | 86.76 | 80.41 |
| Playground v2.5 (Li et al. (2024a)) | 75.47 | 83.06 | 82.59 | 81.20 | 84.08 | 83.50 |
| Hunyuan-DiT (Li et al. (2024b)) | 78.87 | 84.59 | 80.59 | 88.01 | 74.36 | 86.41 |
| DALL-E3 (Betker et al. (2023)) | 83.50 | 90.97 | 89.61 | 88.39 | 90.58 | 89.83 |
| SD3 (Esser et al. (2024)) | 84.08 | 87.90 | 91.01 | 88.83 | 80.70 | 88.68 |
| Playground v3 (Liu et al. (2024)) | 87.04 | 91.94 | 85.71 | 90.90 | 90.00 | 92.72 |
| *Autoregressive models* | | | | | | |
| Emu3-DPO (Wang et al. (2024)) | 81.60 | 87.54 | 87.17 | 86.33 | 90.61 | 89.75 |
| NOVA (0.3B) | 80.60 | 85.41 | 86.97 | 85.16 | 92.05 | 71.20 |
| NOVA (0.6B) | 82.25 | 87.65 | 87.65 | 85.62 | 90.90 | 74.80 |
| NOVA (1.4B) | 83.01 | 86.32 | 88.69 | 86.35 | 91.94 | 74.80 |

## G  MORE TEXT-TO-IMAGE VISUALIZATIONS

We present more text-to-image samples in the Figure 5. NOVA can generate images with a maximum resolution of 1024×1024. Our model excels in the domain of text-to-image generation, producing a vast array of high-quality images that accurately reflect the textual descriptions provided. This capability not only spans a wide range of subjects, from realistic landscapes and portraits to imaginative and abstract concepts, but also maintains a high level of detail and aesthetic quality.

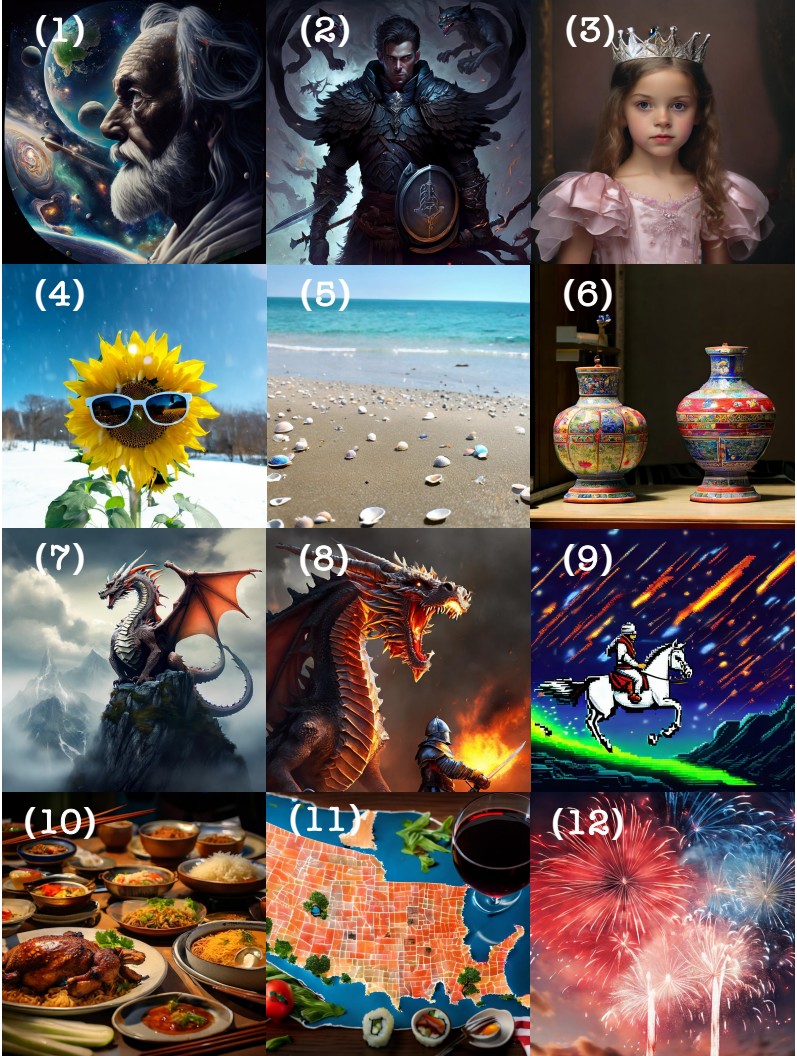

Figure 5: **More text-to-image visualizations.** Text prompts are as follows: (1) "In the foreground is the detailed, head-and-shoulders portrait of an elderly man with a long white beard...", (2) "a digital artwork of a fantasy warrior character. The character is male, depicted from the waist up, and appears to have a stern or serious facial expression...", (3) "a young girl wearing a tiara and frilly dress", (4) "A sunflower in sunglasses dances in the snow", (5) "A beach with no people", (6) "Two Ming vases on the table, the larger one is more colorful than the other", (7) "A dragon perched majestically on a craggy, smoke-wreathed mountain", (8) "a dragon breathing fire onto a knight", (9) "a pixel art style graphic with vibrant colors. It features a single rider on a horse, both depicted in mid-gallop to the left side of the frame...", (10) "A table full of food. There is a plate of chicken rice, a bowl of bak chor mee, and a bowl of laksa", (11) "A map of the United States made out sushi. It is on a table next to a glass of red wine" and (12) "beautiful fireworks in the sky with red, white and blue".

## H  MORE TEXT-TO-VIDEO VISUALIZATIONS

We present more text-to-video samples generated by NOVA in the Figure 6. NOVA can generate videos with a resolution of 33×768×480. Our model stands out in the field of text-to-video generation, capable of producing a substantial number of high-quality videos that vividly bring textual descriptions to life. From detailed storylines and character animations to realistic environmental settings and action scenes, our model demonstrates exceptional proficiency in generating content.

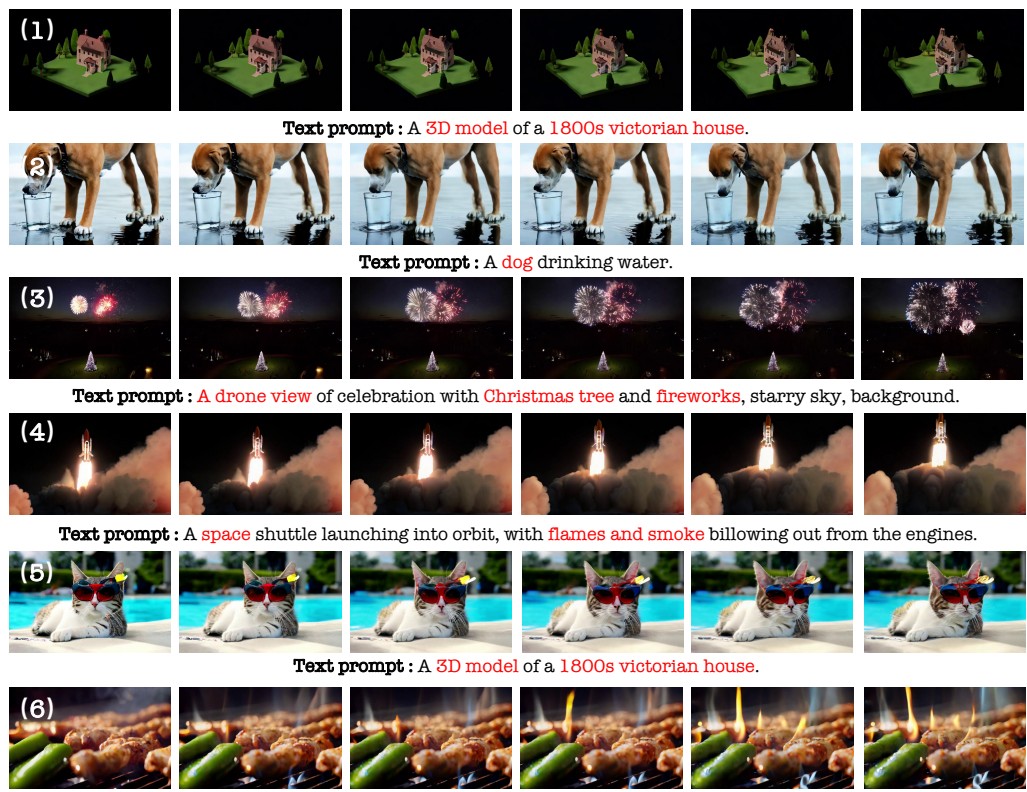

Figure 6: More text-to-video visualizations. Best viewed with zoom for enhanced detail.