# OpenReview forum: "Autoregressive Video Generation without Vector Quantization"
_ICLR.cc/2025/Conference — ICLR 2025 Poster_

### Official Review · Reviewer_xK2A · 2024-10-17

**Soundness:** 3
**Presentation:** 2
**Contribution:** 3
**Rating:** 6
**Confidence:** 3

**Summary:**

This paper proposes a video generation method NOVA, expands MAR[1] from image generation to video generation. However, this process is not smooth sailing. The author encounters problems such as MAR's insufficient ability to model long sequences and poor extrapolation ability. The author proposes a method in the time dimension. Frame-by-frame and set-by-set generated solution strategies in spatial dimensions achieve excellent text-to-image and text-to-video performance.

[1] Tianhong Li, Yonglong Tian, He Li, Mingyang Deng, and Kaiming He. Autoregressive image generation without vector quantization. arXiv preprint arXiv:2406.11838, 2024b.

**Strengths:**

1. As far as I know, it is the first time to expand MAR into the general generation field (text-to-image, text-to-video, etc.), which is a very good attempt.

2. The article achieves excellent text-to-image performance (on T2I CompBench) and stands out from diffusion models. Although I have some doubts about the evaluation setting and comparison methods described in the article, the results are still excellent.

3. As a pre-training paradigm for generative models, especially video generation, this article consumes less resources and is relatively affordable.

**Weaknesses:**

1. The article's explanation of the model architecture is vague and difficult to understand:

    ①The paper mentions temporal encoder, spatial encoder, and decoder with 16 layers each on line 260, but the full article does not explain what the encoder and decoder in spatial layers are.

    ②In Figure 1, when Spatial Layers performs mask modeling on S2, it can directly obtain the complete embedding output by S2 after being encoded by Temporal Layers. Isn't this a kind of information leakage? However, in the actual inference generation At the time of S2, there was no known S2. We only had S1 before we get S2.

2. As far as I know, in the evaluation on T2I-CompBench mentioned by the author in lines 292-295 of the article, there are only 300 evaluation prompts for each category, and the others are training prompts. And according to the setting of the original T2I-CompBench paper, each evaluation prompt should generate 10 images.

3. In the comparison of Table 2, there is no comparison with the most advanced diffusion model. At least it should be compared with SD3 [2] and DALL-E3 [3].

4. For the curve in Figure 8(a), loss is optimized best under the image-to-video (w/o scale&shift) setting. Doesn’t this mean that the scale&shift operation is useless? Then why do we need to add this operation?

[2] Patrick Esser, Sumith Kulal, Andreas Blattmann, Rahim Entezari, Jonas M¨uller, Harry Saini, Yam Levi, Dominik Lorenz, Axel Sauer, Frederic Boesel, et al. Scaling rectified flow transformers for high-resolution image synthesis. In International Conference on Machine Learning, 2024b.

[3] James Betker, Gabriel Goh, Li Jing, Tim Brooks, Jianfeng Wang, Linjie Li, Long Ouyang, Juntang Zhuang, Joyce Lee, Yufei Guo, et al. Improving image generation with better captions. Computer Science. https://cdn.openai.com/papers/dall-e-3.pdf, 2(3):8, 2023.

**Questions:**

In order to ensure fairness when comparing with other methods in Table 2, you should submit the comparison results under the same settings. This is also to ensure that the settings of other methods can be unified when comparing in the future. And you should emphasize whether you conducted the evaluation under zero-shot or the result after fine-tuning on the training set.

---

> ### Author Response · Authors · 2024-11-25
>
> > [**Q1**]. The paper mentions temporal encoder, spatial encoder, and decoder with 16 layers each on line 260, but the full article does not explain what the encoder and decoder in spatial layers are.
>
> [**A1**]. Thank you for your question and valuable feedback. As discussed in Section 3.3 (Line 199), our spatial layers adopt the encoder-decoder architecture of MAR [1], similar to MAE [2]. Specifically, the encoder processes the visible patches for reconstruction. The decoder further processes visible and masked patches for generation. We have revised this part in Sec 4.1.
>
> > [**Q2**]. In Figure 1, when Spatial Layers performs mask modeling on S2, it can directly obtain the complete embedding output by S2 after being encoded by Temporal Layers. Isn't this a kind of information leakage? However, in the actual inference generation At the time of S2, there was no known S2. We only had S1 before we get S2.
>
> [**A2**]. Thanks for your valuable feedback. We acknowledge the ambiguities in Figure 1. To clarify, the input S2′ of the spatial layers originates from the <BOV> token and the S1-attended outputs of the temporal layers, not the <BOV> token and S2-attended outputs. **Thus, there is no information leakage.** We have corrected this in Figure 1 and Section 3.3.
>
> To eliminate any potential confusion, we outline the inference process of NOVA in more detail below:
> - Initially, the output temporal-layer features of the BOV tokens (serving as anchor features) are scaled by 1 and shifted by 0 to act as the first-frame condition in the spatial layers.
> - The first-frame temporal features are projected to scale and shift tensors, which are subsequently used to apply element-wise affine transformations to the anchor features. This process generates the second-frame condition for the spatial layers.
> - During training, this autoregressive process is parallelized using a frame-wise causal mask in the temporal layers.
>
> > [**Q3**]. in the evaluation on T2I-CompBench mentioned by the author in lines 292-295 of the article, there are only 300 evaluation prompts for each category, and the others are training prompts. And according to the setting of the original T2I-CompBench paper, each evaluation prompt should generate 10 images.
>
> [**A3**]. Thank you for pointing out typos. We have thoroughly reviewed our evaluation code and confirmed that all results in the paper were obtained using standard settings: For each sub-category in the testing set, we generate 10 images per prompt for 300 prompts, resulting in a total of 18,000 images generated in a zero-shot manner. We have revised in Sec 4.1.
>
> > [**Q4**]. In the comparison of Table 2, there is no comparison with the most advanced diffusion model. At least it should be compared with SD3 and DALL-E3.
>
> [**A4**]. Thanks for your valuable feedback. Given that SD3 and DALL·E3 were not assessed in T2Ibench, we opted to use Geneval[1] for additional comparisons. By further training NOVA on larger text-to-image datasets, we achieved significantly superior results compared to commercial models like DALL-E 3 and SD3, showcasing its impressive scaling properties as follows:
>
> | Model                     | #Params | Dataset | Overall | Single | Two  | Counting | Colors | Position | Attribution | GPU-days |
> | :-----------------------: | :-----: | :-----: | :-----: | :----: | :--: | :------: | :----: | :------: | :---------: | :------: |
> | DALL-E 3                  |    -    |    -    |  0.67   |  0.96  | 0.87 |   0.47   |  0.83  |   0.43   |    0.45     |    -     |
> | SD3          |   2B    |    -    |  0.62   |  0.98  | 0.74 |   0.63   |  0.67  |   0.34   |    0.36     |    -     |
> | SD3          |   8B    |    -    |  0.68   |  0.98  | 0.84 |   0.66   |  0.74  |   0.40   |    0.43     |    -     |
> | NOVA      |  0.6B   |  16M    |  0.66   |  0.98  | 0.86 |   0.54   |  0.86  |   0.19   |    0.48     |   127    |
> | NOVA                |  0.3B   |  600M   |  0.66   |  0.98  | 0.82 |   0.55   |  0.83  |   0.28   |    0.48     |   135    |
> | NOVA                |  0.6B   |  600M   |  0.69   |  0.99  | 0.88 |   0.60   |  0.86  |   0.29   |    0.52     |   252    |
> | NOVA                |  1.4B   |  600M   |  0.72   |  0.99  | 0.91 |   0.65   |  0.86  |   0.37   |    0.53     |   423    |
>
> [1] Ghosh, Dhruba, Hannaneh Hajishirzi, and Ludwig Schmidt. "Geneval: An object-focused framework for evaluating text-to-image alignment." Advances in Neural Information Processing Systems 36 (2024).

---

> ### Author Response · Authors · 2024-11-25
>
> > [**Q5**]. For the curve in Figure 8(a), loss is optimized best under the image-to-video (w/o scale&shift) setting. Doesn’t this mean that the scale&shift operation is useless? Then why do we need to add this operation?
>
> [**A5**]. Thanks for your comment. As discussed in Lines 428–471, **a lower training loss without the scale-shift layer does not necessarily indicate better generative performance.** We infer that this behavior is caused by **overfitting redundant information among video tokens.** This is clearly demonstrated in Figure 9, which shows that as the inner rank of the MLP in the scale-shift layers increases, NOVA's visual performance declines. Moreover, removing the scale-shift layer altogether results in the worst visual outcomes. By introducing the scale-shift layer and reducing its inner rank, we increase the learning difficulty, which helps NOVA focus more effectively on learning structural and motion priors, as illustrated in Figure 9.
>
> > [**Q6**]. In order to ensure fairness when comparing with other methods in Table 2, you should submit the comparison results under the same settings. This is also to ensure that the settings of other methods can be unified when comparing in the future. And you should emphasize whether you conducted the evaluation under zero-shot or the result after fine-tuning on the training set.
>
> [**A6**]. Thank you for your suggestion. **All results of NOVA presented in this paper were evaluated in a zero-shot setting.** To showcase the generalization and scalability of our model, we trained it on a larger dataset and tested it on multiple benchmarks, as outlined in Q4. We will release the code and model weights to ensure transparency and reproducibility.

---

> ### Author Response · Authors · 2024-11-28
>
> Dear Reviewer xK2A,
>
> Thank you once again for your valuable suggestions and comments. With the discussion deadline approaching, we are more than happy to provide any further clarification you may need.
>
> In our previous response, we took great care in addressing your feedback and have made the following updates and additions:
> - Provided further explanations of our architecture.
> - Offered a detailed description of Fig.1.
> - Provided a revised T2I-CompBench setting.
> - Included a comparison with more SOTA models on GenEval.
> - Provided further explanations of Figure 8(a) and the role of the Scaling and Shift layer.
> - Provided further explanations of evaluation settings.
>
> We trust that the new experiments and explanations have clarified the contributions and enhanced the quality of our submission. If you require any further clarifications or additional analyses, please feel free to reach out.
>
> Thank you for your time and valuable feedback!
>
> Best regards,
> The Authors

---

> ### Author Response · Authors · 2024-12-03
>
> Dear Reviewer xK2A,
>
> Thank you once again for your valuable suggestions and feedback.
>
> As the discussion deadline approaches, please feel free to reach out if you need any further clarifications or additional experiments.

---

### Official Review · Reviewer_yFXn · 2024-10-30

**Soundness:** 2
**Presentation:** 3
**Contribution:** 2
**Rating:** 6
**Confidence:** 4

**Summary:**

The paper introduces NOVA, a novel autoregressive (AR) video generation model that leverages non-quantized tokenizers. NOVA aims to combine the advantages of high-fidelity and high-rate visual compression with in-context learning capabilities, enabling it to integrate multiple generative tasks within a unified framework. The model is designed to factorize AR video generation into temporal frame-by-frame prediction and spatial set-by-set prediction, allowing for efficient and effective video generation. The authors claim that NOVA outperforms existing diffusion models in terms of data efficiency, inference speed, and video fluency, while also demonstrating strong zero-shot generalization across various contexts.

**Strengths:**

1) NOVA's framework is well-structured, combining temporal and spatial autoregressive modeling. This dual approach not only enhances the model's efficiency but also its ability to handle multiple generative tasks within a single model, showcasing the potential for in-context learning.

2) The authors provide a thorough evaluation of NOVA, comparing it with state-of-the-art models across various metrics. The results demonstrate that NOVA not only matches but often surpasses the performance of diffusion models, particularly in terms of data efficiency and inference speed.

**Weaknesses:**

I think the key limitation of this work is the novelty, which seems like an extension of MAR on video generation task.

In Table 3, I don't see improvements in the proposed on the basis of previous diffusion-based methods. Are AR-based methods really needed for video generation tasks? Could the author clarify this?

**Questions:**

See weaknesses.

---

> ### Author Response · Authors · 2024-11-25
>
> > [**Q1**]. seems like an extension of MAR on video generation task.
>
> [**A1**]. Thanks for your feedback.
>
> **(1)** Currently, MAR is limited to class-to-image generation and has not been tested on the more complex task of text-to-image generation. A natural approach to extend MAR to text-to-video generation would involve generating multiple images simultaneously and applying autoregression to a single large image. However, our experiments reveal significant challenges with this method, including issues with mask scheduling, autoregressive setup, computational efficiency, generation quality, and, most importantly, scalability across diverse contexts.
>
> **(2)** While inspired by MAR, notably, **it is non-trivial at all from MAR to NOVA**. Comparing to MAR, NOVA not only addressed the issues stated above, but also presented an efficient and state-of-the-art framework for video generation. **NOVA is a brand-new autoregressive generation framework that first predicts temporal frames sequentially and then processes spatial sets within each frame.** NOVA is also the first to integrate a non-quantized autoregressive model for video generation. This approach improves the model's ability to naturally handle a wide range of contextual conditions, including text-to-image, image-to-video, and video extrapolation, extending beyond the capabilities of existing methods. Moreover, NOVA offers potential compatibility with kv-cache acceleration, ensuring both high efficiency and superior generation quality. We believe that NOVA will bolster community confidence in this emerging technique and inspire further advancements in the field.
>
> > [**Q2**]. In Table 3, I don't see improvements in the proposed on the basis of previous diffusion-based methods.
>
> [**A2**]. Thanks for your valuable comment.
>
> **(1)** We are actively working on experiments involving larger models on extensive video datasets. At present, we have invested considerable effort in training larger text-to-image models using extensive datasets, serving as a stronger foundation for video generation. By further training NOVA on larger text-to-image datasets, we achieved significantly superior results compared to commercial models like DALL-E 3 and SD3, showcasing its impressive scaling properties as follows:
>
> | Model                     | #Params | Dataset | Overall | Single | Two  | Counting | Colors | Position | Attribution | GPU-days |
> | :-----------------------: | :-----: | :-----: | :-----: | :----: | :--: | :------: | :----: | :------: | :---------: | :------: |
> | DALL-E 3                  |    -    |    -    |  0.67   |  0.96  | 0.87 |   0.47   |  0.83  |   0.43   |    0.45     |    -     |
> | SD3          |   2B    |    -    |  0.62   |  0.98  | 0.74 |   0.63   |  0.67  |   0.34   |    0.36     |    -     |
> | SD3          |   8B    |    -    |  0.68   |  0.98  | 0.84 |   0.66   |  0.74  |   0.40   |    0.43     |    -     |
> | NOVA      |  0.6B   |  16M    |  0.66   |  0.98  | 0.86 |   0.54   |  0.86  |   0.19   |    0.48     |   127    |
> | NOVA                |  0.3B   |  600M   |  0.66   |  0.98  | 0.82 |   0.55   |  0.83  |   0.28   |    0.48     |   135    |
> | NOVA                |  0.6B   |  600M   |  0.69   |  0.99  | 0.88 |   0.60   |  0.86  |   0.29   |    0.52     |   252    |
> | NOVA                |  1.4B   |  600M   |  0.72   |  0.99  | 0.91 |   0.65   |  0.86  |   0.37   |    0.53     |   423    |
>
> **(2)** However, these experiments on text-to-video generation are resource-intensive and time-consuming, so obtaining final results will require some time. If we complete them within the review period, we will promptly share the findings.
>
> > [**Q3**]. Are AR-based methods really needed for video generation tasks?
>
> [**A3**]. The temporal autoregressive characteristic enables NOVA to handle a wide range of contextual conditions in one single model, including text-to-image(Figure 3), image-to-video(Figure 4), video extrapolation(Figure 5) and others(Figure 6). Meanwhile, masked spatial-wise autoregression allows for bidirectional modeling combined with per-token denoising process, demonstrating high efficiency(12s vs 50s+ of diffusion models in Table 3) and superior generation quality (75.84 vs 75.66 of opensora 1.1 in Table 3). We believe that NOVA demonstrates strong model capabilities and scaling efficiency, paving a promising route of autoregressive branch for video generation tasks.

---

> ### Author Response · Authors · 2024-11-28
>
> Dear Reviewer yFXn,
>
> Thank you once again for your valuable suggestions and comments. With the discussion deadline approaching, we are more than happy to provide any further clarification you may need.
>
> In our previous response, we took great care in addressing your feedback and have made the following updates and additions:
>
> 1. Provided further explanations of our novelty and contributions to the community.
> 2. Conducted additional experiments to examine the improvements on the basis of previous diffusion-based methods.
> 3. Provided an explanation of the role of AR based methods in video generation task.
>
> We trust that the new experiments and explanations have clarified the contributions and enhanced the quality of our submission. If you require any further clarifications or additional analyses, please feel free to reach out.
>
> Thank you for your time and valuable feedback!
>
> Best regards,
> The Authors

---

> ### Author Response · Authors · 2024-12-03
>
> Dear Reviewer yFXn,
>
> In response to your inquiry about the improvements made to our model based on previous diffusion-based methods, we have gathered an additional 7 million high-quality video samples and retrained our model. This has led to a significant performance improvement, from **75.84 to 80.12**, as shown in the attached table. Our updated results are competitive with the latest state-of-the-art T2V models, such as OpenSoraPlan V1.3 (77.23), LTX-Video (80.00), Mochi-1 (80.13) and CogVideoX-2B (80.91).  These findings further highlight the scalability and robustness of our approach when applied to larger and more complex datasets.
>
> Since the deadline of discussion is approaching, we are happy to provide any additional clarification that you may need.
>
> | Model               | #Params | Dataset | Total Score | A100 days |
> |:------------------:|:-------:|:-------:|:-----------:|:---------:|
> | OpenSoraPlan V1.3   | 2.7B    | 25M     | 77.23       | -         |
> | LTX-Video           | 2B      | -       | 80.00       | -         |
> | Mochi-1             | 10B     | -       | 80.13       | -         |
> | CogVideoX-2B        | 2B      | 35M     | 80.91       | -         |
> | NOVA                | 0.6B    | 13M     | 75.84       | 267       |
> | NOVA                | 0.6B    | 20M     | 80.12       | 342       |

---

### Official Review · Reviewer_nKGc · 2024-11-02

**Soundness:** 3
**Presentation:** 3
**Contribution:** 3
**Rating:** 6
**Confidence:** 4

**Summary:**

This paper proposes a video generation framework called NOVA, which is based on autoregressive modeling. NOVA performs temporal frame-by-frame prediction and spatial set-by-set prediction. This approach leverages the in-context learning and extrapolation advantages of autoregressive models while maintaining the efficiency of bidirectional modeling. Compared to existing video generative models, NOVA achieves higher data efficiency, faster inference speeds, and about similar video generation quality with fewer parameters.

**Strengths:**

* NOVA achieves state-of-the-art (SOTA) results in text-to-image (T2I) tasks.
* NOVA shows much faster inference speeds than previous video generative models.
* The method of combining temporal autoregressive and spatial bidirectional modeling is simple yet effective.
* The Scaling and Shift Layer is also simple but effective. Also, the analysis of the layer is comprehensive.

**Weaknesses:**

* While NOVA achieves SOTA in T2I, this aspect feels like a straightforward extension of MAR[1] rather than a novel contribution.
* For text-to-video (T2V), NOVA uses relatively less data and fewer parameters and has fast inference speeds but falls short in performance. Therefore, it needs further testing about scalability (i.e., if NOVA can match the performance of the open-source models in the main table when scaled up.).
* There is a question about whether extrapolation is truly unique to autoregressive (AR) models. Diffusion models and bidirectional models could also potentially achieve extrapolation through a sliding window approach, which would need a comparative analysis.
* The ablation study on frame-by-frame autoregressive modeling lacks clarity, which is critical given the importance of this topic to the authors' main arguments. The qualitative results in Figure 7. appear less convincing when viewed in images, and it’s unclear if NOVA did not have similar subtle limitations. A side-by-side comparison with the same text prompt or inclusion of video examples would be helpful.

In short, \
Although the methods presented are simple and novel, the primary claims are not clearly backed by the experiments. The main claims are: (1) NOVA combines the advantages of temporal autoregressive modeling with spatial bidirectional modeling, and (2) NOVA demonstrates data and parameter efficiency in practice. However, these claims are not sufficiently supported by the experimental results.

If the authors provide clearer support for their claims with additional analysis, I would be happy to raise my score.

[1] Li, et al. "Autoregressive Image Generation without Vector Quantization." arXiv 2024.

**Questions:**

* Suggestion: To make the Scaling and Shift Layer easier to understand, the authors could mention its similarity to FiLM[1] or AdaIN[2].
* Typo: Line 296 - each with

[1] Perez, et al. "Film: Visual reasoning with a general conditioning layer." AAAI 2018.\
[2] Huang, Xun, and Serge Belongie. "Arbitrary style transfer in real-time with adaptive instance normalization." ICCV 2017.

**Details Of Ethics Concerns:**

.

---

> ### Author Response · Authors · 2024-11-25
>
> > [**Q1**]. feels like a straightforward extension of MAR
>
> [**A1**]. Thanks for your feedback.
>
> **(1)** Currently, MAR is limited to class-to-image generation and has not been tested on the more complex task of text-to-image generation. A natural approach to extend MAR to text-to-video generation would involve generating multiple images simultaneously and applying autoregression to a single large image. However, our experiments reveal significant challenges with this method, including issues with mask scheduling, autoregressive setup, computational efficiency, generation quality, and, most importantly, scalability across diverse contexts.
>
> **(2)** While inspired by MAR, notably, **it is non-trivial at all from MAR to NOVA**. Comparing to MAR, NOVA not only addressed the issues stated above, but also presented an efficient and state-of-the-art framework for video generation. **NOVA is a brand-new autoregressive generation framework that first predicts temporal frames sequentially and then processes spatial sets within each frame.** NOVA is also the first to integrate a non-quantized autoregressive model for video generation. This approach improves the model's ability to naturally handle a wide range of contextual conditions, including text-to-image, image-to-video, and video extrapolation, extending beyond the capabilities of existing methods. Moreover, NOVA offers potential compatibility with kv-cache acceleration, ensuring both high efficiency and superior generation quality. We believe that NOVA will bolster community confidence in this emerging technique and inspire further advancements in the field.
>
> > [**Q2**]. testing about scalability
>
> [**A2**]. Thanks for your valuable comment.
>
> **(1)** We are actively working on experiments involving larger models on extensive video datasets. At present, we have invested considerable effort in training larger text-to-image models using extensive datasets, serving as a stronger foundation for video generation. By further training NOVA on larger text-to-image datasets, we achieved significantly superior results compared to commercial models like DALL-E 3 and SD3, showcasing its impressive scaling properties as follows:
>
> | Model                     | #Params | Dataset | Overall | Single | Two  | Counting | Colors | Position | Attribution | GPU-days |
> | :-----------------------: | :-----: | :-----: | :-----: | :----: | :--: | :------: | :----: | :------: | :---------: | :------: |
> | DALL-E 3                  |    -    |    -    |  0.67   |  0.96  | 0.87 |   0.47   |  0.83  |   0.43   |    0.45     |    -     |
> | SD3          |   2B    |    -    |  0.62   |  0.98  | 0.74 |   0.63   |  0.67  |   0.34   |    0.36     |    -     |
> | SD3          |   8B    |    -    |  0.68   |  0.98  | 0.84 |   0.66   |  0.74  |   0.40   |    0.43     |    -     |
> | NOVA      |  0.6B   |  16M    |  0.66   |  0.98  | 0.86 |   0.54   |  0.86  |   0.19   |    0.48     |   127    |
> | NOVA                |  0.3B   |  600M   |  0.66   |  0.98  | 0.82 |   0.55   |  0.83  |   0.28   |    0.48     |   135    |
> | NOVA                |  0.6B   |  600M   |  0.69   |  0.99  | 0.88 |   0.60   |  0.86  |   0.29   |    0.52     |   252    |
> | NOVA                |  1.4B   |  600M   |  0.72   |  0.99  | 0.91 |   0.65   |  0.86  |   0.37   |    0.53     |   423    |
>
> **(2)** However, these experiments on text-to-video generation are resource-intensive and time-consuming, so obtaining final results will require some time. If we complete them within the review period, we will promptly share the findings.
>
> > [**Q3**]. whether extrapolation is truly unique to autoregressive (AR) models.
>
> [**A3**]. Thanks for your comment. We acknowledge that several training-free methods [1] enable text-to-video diffusion models to extrapolate beyond their training lengths. However, these approaches often struggle with temporal inconsistencies [2]. A more practical alternative is to train (or fine-tune) a separate video-to-video diffusion model specifically for video extrapolation. In contrast, NOVA's autoregressive design allows it to naturally generalize across extended video durations and, more importantly, handle a diverse range of zero-shot tasks within a single unified model. Its temporal autoregressive framework enables seamless execution of various tasks, such as text-to-image, text+image-to-video, and text+video-to-video generation, offering significantly more flexibility compared to diffusion-based methods.
>
> [1] Ho, J., Salimans, T., Gritsenko, A., Chan, W., Norouzi, M., & Fleet, D. J. (2022). Video Diffusion Models. http://arxiv.org/abs/2204.03458
>
> [2] Luo, Z., Chen, D., Zhang, Y., Huang, Y., Wang, L., Shen, Y., Zhao, D., Zhou, J., & Tan, T. (2023). VideoFusion: Decomposed Diffusion Models for High-Quality Video Generation. 10209–10218. http://arxiv.org/abs/2303.08320

---

> ### Author Response · Authors · 2024-11-25
>
> > [**Q4**]. The ablation study on frame-by-frame autoregressive modeling lacks clarity
>
> [**A4**]. Thanks for your comment. For better convenience and clearer presentation, we have refined figure 7 in the revised version. In addition, we have added a comparison of quantitative results in Section E of the supplementary material. Our findings are summarized as follows:
>
> **(1) Efficient Motion Modeling** : We observed that the total score marginally lower than compared to NOVA(75.38 vs 75.84), especially in terms of the dynamic degree metric, which showed a more pronounced decline(11.38 vs. 23.27). We hypothesize that while bidirectional attention enhances model capacity, it demands more extensive data and longer training times to capture subtle motion changes compared to causal models.
>
> **(2) Efficient Video Inference** : Thanks to the autoregressive generation of the kv-cache technology and frame-by-frame processing, NOVA's inference time is much faster compared to methods without TAM, with a greater speed advantage for longer videos.
>
> | Model               | Total Score | Dynamic Degree | Infer Time |
> | :------------------ | :---------: | :------------: | :--------: |
> | NOVA                | 75.84       |     23.27      |     12s    |
> | NOVA (w/o TAM)      | 75.38       |     11.38      |     39s    |
>
> > [**Q5**]. However, these claims are not sufficiently supported by the experimental results.
>
> [**A5**]. Thank you for your question and valuable feedback.
>
> **claim 1 :  NOVA combines the advantages of temporal autoregressive modeling with spatial bidirectional modeling.**
> The temporal autoregressive characteristic enables NOVA to handle a wide range of contextual conditions in one single model, including text-to-image(Figure 3), image-to-video(Figure 4), video extrapolation(Figure 5) and others(Figure 6).
> Meanwhile, masked spatial-wise autoregression allows for bidirectional modeling combined with per-token denoising process, demonstrating high efficiency(12s vs 50s+ of diffusion models in Table 3) and superior generation quality (0.72 vs 0.68 of SD3 in Supp. Table 2). We believe that these results can well validate that NOVA combines the superiority of both AR and diffusion approaches.
>
> **claim 2 :  NOVA demonstrates data and parameter efficiency in practice.**
> With fewer computational resources and model capacity, NOVA achieves notable results in text-to-image generation (0.4955 vs 0.4815 of PixArt-α in Table 2) and comparable performance in text-to-video domain (75.84 vs 75.66 of opensora 1.1 in Table 3). In the supplementary material, we present additional text-to-image experiments using more data and larger model capacities. As shown in Supp. Table 2, NOVA with 1.4B parameters achieves state-of-the-art (SOTA) results (0.72 vs. 0.68 for the strongest SD3), highlighting its promising potential for text-to-video generation. If additional experiments are completed within the review period, we will promptly share the findings. These results underscore NOVA’s strong model capabilities and scaling efficiency, demonstrating notable data and parameter efficiency.
>
> > [**Q6**]. To make the Scaling and Shift Layer easier to understand, the authors could mention its similarity to FiLM[ or AdaIN.
>
> [**A6**]. Thank you for your suggestions. We have supplemented the description of Scaling and Shift Layer in the revised appendix for better understanding.
>
> > [**Q7**]. Typo: Line 296 - each with
>
> [**A6**]. Thanks for your reminder, we have corrected it in the revised

---

> ### Author Response · Authors · 2024-11-28
>
> Dear Reviewer nKGc,
>
> Thank you once again for your valuable suggestions and comments. With the discussion deadline approaching, we are more than happy to provide any further clarification you may need.
>
> In our previous response, we took great care in addressing your feedback and have made the following updates and additions:
>
> 1. Provided further explanations of our novelty and contributions to the community.
> 2. Included a comparison of the scaling behavior of NOVA.
> 3. Provided an explanation of the relationship between extrapolation and autoregressive (AR) Models.
> 4. Conducted additional experiments to examine the effect of frame-by-frame autoregressive modeling.
> 5. Provided detailed explanations of our two claims.
> 6. Provided a detailed description of the Scaling and Shift Layer.
> 7. Fixed some typos.
>
> We trust that the new experiments and explanations have clarified the contributions and enhanced the quality of our submission. If you require any further clarifications or additional analyses, please feel free to reach out.
>
> Thank you for your time and valuable feedback!
>
> Best regards,
> The Authors

---

> ### Comment · Reviewer_nKGc · 2024-12-01
>
> First, I’d like to thank the authors for sincerely addressing my comments and conducting the experiments.
>
> After reviewing the authors' responses, my concerns regarding the novelty, scalability, and the effectiveness of TAM have been resolved. While it would have been nice to see scalability experiments for NOVA on video data in addition to images, I believe the NOVA's temporal autoregressive modeling and the expansion of a Transformer without vector quantization to large-scale text-to-image and text-to-video data have significant value. Therefore, I will raise my score from 5 to 6.
>
> This paper is already valuable as it is, and while I understand the cost challenges of conducting video experiments, I believe it would become an even stronger paper if scalability experiments on videos were added in the future.

---

> > ### Author Response · Authors · 2024-12-03
> >
> > Dear Reviewer nKGc,
> >
> > Thank you for your positive feedback and for increasing your score. We are pleased to hear that our responses have addressed your concerns regarding the novelty, scalability, and effectiveness of our model.
> >
> > In response to your suggestion about scalability experiments with video data, we have collected an additional 7 million high-quality video samples and retrained our model. This has led to a significant performance improvement, from **75.84 to 80.12**, as shown in the attached table. Our updated results are competitive with the latest state-of-the-art T2V models, such as OpenSoraPlan V1.3 (77.23), LTX-Video (80.00), Mochi-1 (80.13) and CogVideoX-2B (80.91). These findings further highlight the scalability and robustness of our approach when applied to larger and more complex datasets.
> >
> > We sincerely appreciate your valuable suggestions and continued support. Thank you once again.
> >
> > | Model               | #Params | Dataset | Total Score | A100 days |
> > |:------------------:|:-------:|:-------:|:-----------:|:---------:|
> > | OpenSoraPlan V1.3   | 2.7B    | 25M     | 77.23       | -         |
> > | LTX-Video           | 2B      | -       | 80.00       | -         |
> > | Mochi-1             | 10B     | -       | 80.13       | -         |
> > | CogVideoX-2B        | 2B      | 35M     | 80.91       | -         |
> > | NOVA                | 0.6B    | 13M     | 75.84       | 267       |
> > | NOVA                | 0.6B    | 20M     | 80.12       | 342       |

---

### Official Review · Reviewer_GfSG · 2024-11-04

**Soundness:** 3
**Presentation:** 3
**Contribution:** 4
**Rating:** 8
**Confidence:** 4

**Summary:**

This paper introduces an autoregressive model combined with a diffusion model as the prediction head, enabling video generation without vector quantization. The proposed approach, termed NOVA, maintains the causal property of AR models' temporal frame-by-frame prediction, while leveraging bidirectional modeling within individual frames (spatial set-by-set prediction).  NOVA achieves state-of-the-art text-to-image and text-to-video generation performance with significantly lower training costs and higher inference speed.

**Strengths:**

- As a follower of MAR (Tianhong Li et al. (2024b)), this paper for the first time lifts the non-quantized AR model to video generation. In contrast to trivially modifying the 2D non-quantized MAR to a 3D version, they design the autoregressive modeling sequentially that integrates first temporal frame-by-frame prediction and then spatial set-by-set within each frame. This facilitates the model's ability of video extrapolation and potential  compatibility with kv-cache acceleration.
- The model is trained on large-scale text-to-image and text-to-video datasets (trained from scratch) and shows high image and video generation quality compared to existing SOTA models. It will have a great potential contribution to the vision community If the pre-trained checkpoint is released.

- This paper provides valuable empirical design spirits as vilified by their experiments:
  - Instead of directly assigning the current frame of temporal layers’ outputs to the spatial layer as indicator features (for predicting the next frame), they propose to use the BOV-attended output as an anchor feature and inject the current frame's output via the Scale & Shift of LayerNorm. This technique improves the training stability and alleviates cumulative inference errors. It is a valuable prior (design choice) for subsequent studies on autoregressive long video generation.
  - They conducted extensive ablation studies to show using post-norm layers before the residual connections is a better design choice for a smoother and more stable training process.

Tianhong Li, Yonglong Tian, He Li, Mingyang Deng, and Kaiming He. Autoregressive image generation without vector quantization. arXiv preprint arXiv:2406.11838, 2024b.

**Weaknesses:**

- Unclear training/inference details.
  1. According to Figure 1. At training time, the model predicts a set of masked tokens of the 2nd frame. At inference time, the model progressively reduces the masked ratio from 1.0 to 0. However, as the 1st and 2nd frames have been generated (as the given conditional frames) in the Fig.1's example, the model should progressively unmask the 3rd frame. There seems to be some inconsistency between training and inference. In other words, for the example in Fig.1, which frame is modeled by $x_n^t$  ?
  2. It might be helpful to provide a step-by-step explanation of how frames are generated during inference, particularly highlighting any differences from training.

- Unclear video extrapolation setting.

  1. The exact number of frames used during training is not clear. According to all the information available in the paper, the model is trained on samples with 29 frames (Line315: 2.4s x 12 FPS = 28.8 frames. Line 296: the model generates 29 frames for evaluation). But this is not clearly evidenced in the descriptions of training details.

  2. It's unclear how context is handled when generating videos longer than the training length. Suppose that the model was trained on 29 frames. Is the context truncated when the length of video extrapolation exceeds 29 frames? For example, from my perspective, the video extrapolation is like:

    ```
    ...
    [x_1,...,x_28]--> x_29 # training length is reached
    [x_2,...,x_29]--> x_30  # earliest context (x_1) is truncated
    ...
    ```

  3. How are the 1D sine-cosine temporal positional embeddings applied for frames beyond the training length? This information would help clarify the model's capabilities and limitations in video extrapolation.

**Questions:**

- From my understanding, the generation of the next frame is achieved by one step of temporal autoregression (stage-1) followed by several steps of spatial autoregression (i.e.,  progressively reducing the masks from 1.0 to 0) (stage-2).  What's the time cost for each of these two stages?
- As claimed in the paper, the Scaling and Shift Layer effectively reduces the cumulative inference errors. What is the limit of the model's video extrapolation capability? In other words, after how many autoregression steps will the cumulative errors severely affect the frame quality?
  - A quantitative metric on frame quality vs. the number of autoregression steps might be helpful. Or some qualitative examples that show quality degradation in long-term autoregressive generation.
- Suggestions:
  - In Figure 7, better to provide the results of NOVA and compare them with the results of the simple baseline.
  - Better to add frame ids in the qualitative examples.

---

> ### Author Response · Authors · 2024-11-25
>
> > [**Q1**]. inconsistency between training and inference
>
> [**A1**]. Thanks for your valuable feedback. We acknowledge the ambiguities in Figure 1. To clarify, the input S2′ of the spatial layers originates from the <BOV> token and the S1-attended outputs of the temporal layers, not the <BOV> token and S2-attended outputs. Thus, **there is no discrepancy between the training and inference phases of NOVA**.
>
> > [**Q2**]. A step-by-step explanation of how frames are generated during inference.
>
> [**A2**]. Specifically, the temporal-layer outputs of <BOV> tokens (which also serve as anchor features) are scaled by 1 and shifted by 0 to act as the first-frame condition in the spatial layers. The first-frame temporal features are further projected into scale and shift tensors, which are applied via element-wise affine transformations to the anchor features, forming the second-frame condition in the spatial layers. During training, this autoregressive process is parallelized using a frame-wise causal mask in the temporal layers. We recognize that the misrepresentation in Figure 1 regarding the connection between temporal and spatial layers contributed to this misunderstanding. We have corrected this in Figure 1 and Section 3.3.
>
> > [**Q3**]. The exact number of frames used during training is not clear.
>
> [**A3**]. NOVA processes 29 frames during both training and testing processes, corresponding to approximately 2.417 seconds at a frame rate of 12 frames per second. To ensure clarity, we have refined the related descriptions to avoid any ambiguity.
>
> > [**Q4**]. Is the context truncated when the length of video extrapolation exceeds 29 frames?
>
> [**A4**]. Thank you for your comment.
>
> **(1) In our approach, we perform video extrapolation by inferring frames within each training-length clip**, which is consistent with the training process. For example, to generate the 30th frame, we condition on the text, flow, BOV tokens, and the 29th frame, instead of relying on all preceding frames.
> ```
> [text, flow, bov, x_1, ..., x_28] -> x_29 # training length is reached
> [text, flow, bov, x_29] -> x_30 # infer video extrapolation frames within a new training length
> ...
> [text, flow, bov, x_29, ..., x_56] -> x_57 # a new training length is reached
> ...
> [text, flow, bov, x_57] -> x_58 # infer video extrapolation frames within a next training length
> ```
> This approach utilizes an anchor feature set shared across all generated frames, ensuring coherence in extrapolated video frames. We found that this straightforward strategy effectively maintains temporal consistency while significantly reducing the computational cost of extrapolation.
>
> **(2) We also explore inferring video extrapolation frames within a fixed-length video clip.** However, the use of 1D temporal positional embeddings requires recalculating the mixed features of positional and previous frame features during each extrapolation, which affects computational efficiency. In the future, we plan to introduce 1D-ROPE[1] alternatives to improve compatibility with this approach and enable a more flexible video extrapolation process.
>
> [1] Su, Jianlin, et al. "Roformer: Enhanced transformer with rotary position embedding." Neurocomputing 568 (2024): 127063.
>
> > [**Q5**]. How are the 1D sine-cosine temporal positional embeddings applied for frames beyond the training length?
>
> [**A5**]. Thank you for your comment. As described earlier, we perform video extrapolation by predicting frames within each training-length clip. In this process, a fixed number of frames is generated from scratch for each segment, mirroring the training setup. This approach ensures consistency with the training phase while allowing efficient reuse of features from previous frames through the KV-cache. In the future, we plan to explore alternatives to 1D-ROPE to enhance compatibility with fixed-length window strategies, paving the way for a more flexible and efficient video extrapolation process.

---

> ### Author Response · Authors · 2024-11-25
>
> > [**Q6**]. What's the time cost for each stage of temporal and  spatial autoregression?
>
> [**A6**]. We report inference times on a single NVIDIA A100 GPU (40GB) with a batch size of 24. The input video resolution is 29 × 768 × 480. Temporal layers account for only 0.03 seconds, while spatial layers take 11.97 seconds, highlighting the exceptional efficiency of the temporal layers. Notably, NOVA demonstrates high efficiency in text-to-video generation and offers potential for further acceleration. Techniques such as speculative decoding [1] present promising avenues for future optimization, which we plan to explore.
>
> | Resolution  | Temporal Layers Time | Spatial Layers Time | All Time  |
> | :---------: | :------------------: | :-----------------: | :-------: |
> | 768×480×29  |          0.03s       |        11.97s       |    12s    |
>
> [1] Wang, Zili, et al. "Continuous Speculative Decoding for Autoregressive Image Generation." arXiv preprint arXiv:2411.11925 (2024).
>
> > [**Q7**]. What is the limit of the model's video extrapolation capability?
>
> [**A7**]. Thank you for your valuable feedback. Through visualization experiments, we empirically found that the model can extrapolate up to **3$\times$training length**. We conducted quantitative experiments using LPIPS and PSNR metrics, and discovered that per-frame PSNR values decrease, while LPIPS scores increase over time. Nevertheless, the generated frames show a high degree of similarity to the original video in terms of both content and image quality. For detailed experimental settings and discussions, see Section C of the supplementary material.
>
> > [**Q8**]. better to provide the results of NOVA and compare them with the results of the simple baseline.
>
> [**A8**]. Thank you for your valuable suggestion. We have completed the revisions in the revised version.

---

> > ### Comment · Reviewer_GfSG · 2024-12-03
> >
> > I'd like to appreciate the authors' response and clarifications. My concerns and confusions are resolved. I will keep my score as accept.
> >
> > For Q4 & A4, I still have a  minor question: As stated in A4-(1) and A4-(2), using (2) "inferring video extrapolation frames within a fixed-length video clip", i.e.,
> > ```
> > [text, flow, bov, x_1, ..., x_28] -> x_29 # training length is reached
> > [text, flow, bov, x_2, ..., x_29] -> x_30 # infer video extrapolation frames
> > ```
> >  enables the model to acquire more context.
> >
> > Will (2) achieve better quality (e.g., in terms of temporal consistency) than (1)? (at the cost of redundant calculation).

---

> > > ### Author Response · Authors · 2024-12-03
> > >
> > > Dear Reviewer GfSG,
> > >
> > > Thank you for your positive feedback and for resolving your concerns.
> > >
> > > Regarding your question, **yes**, (2) does achieve better quality in terms of temporal consistency and fewer artifacts, at the cost of redundant calculations. The richer context provided to the model leads to more stable motion and improved overall frame quality.
> > >
> > > We appreciate your insightful comment.

---

> ### Author Response · Authors · 2024-11-28
>
> Dear Reviewer GfSG,
>
> We express our sincere gratitude to the reviewer for dedicating time to review our paper. In our previous response, we took great care in addressing your feedback and have made the following updates and additions:
>
> 1. Offered a detailed description of Fig.1.
> 2. Offered a detailed explanation of frame generation during inference.
> 3. Offered an unambiguous description of frame numbers.
> 4. Offered a detailed video extrapolation process.
> 5. Conducted additional inference time analysis of temporal and spatial layers.
> 6. Conducted additional experiments to evaluate the limit of the model's video extrapolation capability.
> 7. Adjustments made to make the visualization results more clearly displayed
>
> We trust that the new experiments and explanations have clarified the contributions and enhanced the quality of our submission. If you require any further clarifications or additional analyses, please feel free to reach out.
>
> Thank you for your time and valuable feedback!
>
> Best regards,
>
> The Authors

---

### Author Response · Authors · 2024-11-28
**General Response: Contributions and New Experiments**

We sincerely thank all the reviewers for their thoughtful comments and constructive suggestions, which have greatly helped strengthen our work. In this response, we address specific reviewer feedback and also highlight the novel contributions of our work, as well as new experiments we have included in the rebuttal.


*Contributions*
1. **NOVA is the first non-quantized autoregressive model for video generation.** NOVA is a brand-new autoregressive generation framework that first predicts temporal frames sequentially and then processes spatial sets within each frame.
2. **NOVA demonstrates remarkably high efficiency and state-of-the-art results in text-to-image generation and text-to-video generation.** For instance, NOVA demonstrates minimal inference costs, requiring only 12s compared to 50+s for existing diffusion models. With a larger training dataset, it achieves a superior score of 0.72 v.s. 0.68 for commercial SD3 on Geneval, showcasing better scaling efficiency and greater potential. Notably, our original T2V version also delivers higher video generation quality, achieving 75.84 compared to OpenSora's 75.66 on VBench. Moreover, it seamlessly incorporates various conditioning contexts within a single model, significantly underscoring its impact on advancing the field of autoregressive video generation.
3. **NOVA paves the way for next-generation video generation and world models.** It offers valuable insights and possibilities for real-time and infinite video generation, going beyond Sora-like video diffusion models.

*Additional Experiments*

In response to reviewers' suggestions, we have added several new experiments to further support our findings. Below is a summary of the additional experiments included in the rebuttal:
- Video extrapolation evaluations [GfSG]
- Inference time analysis [GfSG]
- Ablations on the impact of temporal autoregressive modeling [nKGc]
- Comparison with leading text-to-image methods on GenEval  [nKGc, yFXn, xK2A]
- Comparison with leading text-to-Video methods on Vbench  [nKGc, yFXn]

Thank you again for your time and efforts. We look forward to your feedback.

---

### Meta-Review · Area_Chair_R9aR · 2024-12-18

**Metareview:**

(a) The paper introduces NOVA, a novel autoregressive framework for video generation without vector quantization. It combines temporal frame-by-frame prediction with spatial set-by-set modeling. NOVA achieves state-of-the-art inference speed, data efficiency, and competitive text-to-image and text-to-video quality.

(b) Strengths include efficiency, superior scaling potential, reduced computational cost, and robust generalization to zero-shot tasks.

(c) Weaknesses involve unclear scalability on larger video datasets and novelty concerns as an extension of MAR. More detailed comparisons with diffusion methods would strengthen claims.

(d) Decision: Accept. The paper offers significant contributions to video generation and addresses reviewer concerns with clarity and additional experiments.

**Additional Comments On Reviewer Discussion:**

During the rebuttal, reviewers questioned scalability, architectural clarity, and NOVA's novelty over MAR. The authors provided detailed clarifications, added experiments for video extrapolation, clarified inference processes, and demonstrated improved performance on larger datasets. These updates addressed concerns and reinforced NOVA's contributions, leading to a favorable decision.

---

### Decision · Program_Chairs · 2025-01-22

Accept (Poster)